# Egr2 and 3 control inflammation, but maintain homeostasis, of PD-1[high] memory phenotype CD4 T cells

Alistair LJ Symonds[1],*, Wei Zheng[2],*, Tizong Miao[1], Haiyu Wang[2], TieShang Wang[2], Ruth Kiome[3], Xiujuan Hou[2], Suling Li[3], Ping Wang[1]

**The transcription factors Egr2 and 3 are essential for controlling inflammatory autoimmune responses of memory phenotype (MP) CD4 T cells. However, the mechanism is still unclear. We have now found that the Egr2+ subset (PD-1high MP) of MP CD4 T cells expresses high levels of checkpoint molecules (PD-1 and Lag3) and also markers of effector T cells (CXCR3 and ICAM-1). Egr2/3 are not required for PD-1high MP CD4 cell development but mediate a unique transcriptional programme that effectively controls their inflammatory responses, while promoting homeostatic proliferation and adaptive responses. Egr2 negative PD-1high MP CD4 T cells are impaired in homeostatic proliferation and adaptive responses against viral infection but display inflammatory responses to innate stimulation such as IL-12. PD-1high MP CD4 T cells have recently been implicated in rheumatoid arthritis pathogenesis, and we have now found that Egr2 expression is reduced in PD-1high MP CD4 T cells from patients with active rheumatoid arthritis compared with healthy controls. These findings demonstrate that Egr2/3 control the inflammatory responses of PD-1high MP CD4 T cells and maintain their adaptive immune fitness.**

## Introduction

Checkpoint molecules such as PD-1 and Lag3 on T cells are important for the control of autoimmune pathology (Zhang & Vignali, 2016). Antigen persistence, such as in chronic infections and tumours, can induce PD-1 and Lag3 expression which can lead to exhaustion of effector T cells (Wherry, 2011). In addition to its role in exhaustion, PD-1 is expressed in memory phenotype (MP), but not naïve, CD4 T cells in the steady state and plays an important role in peripheral tolerance and the prevention of autoimmunity in mouse models (Lin et al, 2007; Thangavelu et al, 2011; Pauken et al, 2015). Lag3 is also expressed in MP CD4 T cells and is involved in regulation of homeostasis (Nakachi et al, 2017). However, despite the suppressive function of the PD-1–PD-L1 pathway on TCR-mediated proliferation, recently it has been discovered that PD-1[high] MP CD4 T cells are pathogenic in Rheumatoid Arthritis (RA) and systemic lupus erythematosus (SLE) patients and are not only inflammatory but also promote the responses of autoimmune B cells (Rao et al, 2017; Bocharnikov et al, 2019; Caielli et al, 2019; Zhang et al, 2019), indicating that regulatory mechanisms in these cells control their homeostasis in the steady state.

The transcription factors Egr2 and 3 are expressed in MP CD4 T cells in the steady state and defects in these two molecules in T cells lead to inflammatory activation and the development of autoimmune symptoms (Zhu et al, 2008; Li et al, 2012; Morita et al, 2016). Although they were initially implicated in inhibition of T-cell proliferation (Harris et al, 2004; Safford et al, 2005), Egr2/3 are not generic inhibitors of T-cell proliferation but are required for clonal expansion of effector T cells in response to viral infection (Miao et al, 2017). Furthermore, Egr2 and 3 do not directly inhibit the proliferation of tolerant T cells, but effectively control inflammatory responses of both effector and tolerant T cells (Omodho et al, 2018). We found that Egr2/3 are only expressed in a subset of MP CD4 T cells, but the phenotypes and function of Egr2/3 expressing MP CD4 T cells are largely unknown.

Here, we show that Egr2/3 are stably expressed in a subset of MP CD4 T cells which express high levels of PD-1 and Lag3 (PD-1[high] MP CD4 T cells) as well as activation markers. Egr2/3 are not required for the development of PD-1[high] MP CD4 T cells but instead are essential for their homeostatic proliferation as well as control of their inflammatory function in the steady state. These functions of Egr2/3 in PD-1[high] MP CD4 T cells are required for the maintenance of a diverse repertoire of MP T cells, which is important for adaptive responses against viral infection. Egr2 regulates the expression of genes in PD-1[high] MP CD4 T cells involved in proliferation, metabolism, and homeostasis as well as inflammation. In the absence of Egr2 and 3, PD-1[high] MP CD4 T cells displayed impaired homeostatic proliferation and adaptive responses but skewed TCR repertoires and innate-like inflammatory function. We also found that Egr2 is

[1]The Blizard Institute, Barts and The London School of Medicine and Dentistry, Queen Mary University of London, London, UK   [2]Division of Rheumatology, Dong Fang Hospital, Beijing University of Chinese Medicine, Beijing, China   [3]Bioscience, Brunel University, Uxbridge, UK

Correspondence: houxiujuan2008@163.com; su-ling.li@brunel.ac.uk; p.wang@qmul.ac.uk
*Alistair LJ Symonds and Wei Zheng contributed equally to this work

expressed in PD-1[high] MP CD4 T cells in human peripheral blood and its expression is impaired in patients with active RA. Thus, the homeostasis of PD-1[high] MP CD4 T cells, regulated by Egr2/3, is important for both the control of inflammatory autoimmune diseases and efficient adaptive immune responses.

# Results

### The transcription factors Egr2 and 3 are stably expressed in a subset of MP CD4 T cells

Egr2/3 have been found to be expressed in MP CD4 T cells (Zhu et al, 2008; Li et al, 2012). In mice with defects in Egr2/3 in T cells, MP CD4 T cells accumulate and are inflammatory (Li et al, 2012). However, the phenotype of Egr2/3 expressing MP CD4 T cells in the steady state is unknown. We found that only a subset of MP T cells expressed Egr2 (Fig 1A). We analysed the phenotype of Egr2[+] and Egr2[−] MP CD4 T cells and found that Egr2[+] MP CD4 T cells expressed high levels of the checkpoint molecules PD-1 and also Lag3, as well as markers associated with effector-like T cells (CCR5, CXCR3, and ICAM-1) (Fig 1B). We term these cells as PD-1[+] or PD-1[high] MP CD4 T cells. To determine the stability of Egr2 expression in these cells, naïve T cells and Egr2[−] and Egr2[+] MP T cells were isolated from GFP-Egr2 knock-in mice (CD45.2) and adoptively transferred into separate wild-type mice (CD45.1). 3 wk after transfer, naïve T cells and Egr2[−] MP T cells remained Egr2[−], whereas Egr2[+] MP T cells largely retained Egr2 expression (Fig 1C and D). Therefore, in contrast to transient expression in effector T cells in response to viral infection (Miao et al, 2017), Egr2 expression is maintained in PD-1[high] MP CD4 T cells.

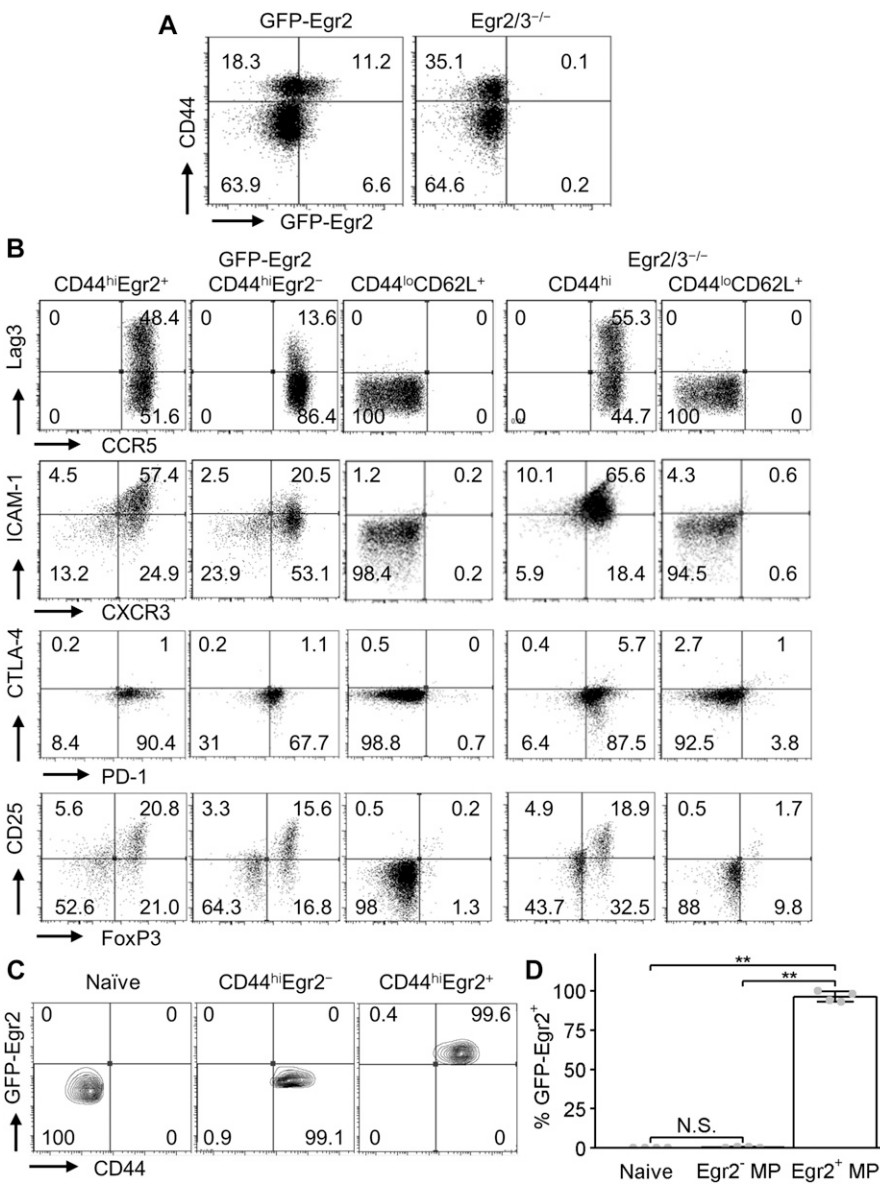

**Figure 1.  Egr2 expression is maintained in a subset of memory phenotype (MP) CD4 T cells.**
**(A)** CD44 and GFP-Egr2 expression in gated CD4 T cells from spleens and lymph nodes of GFP-Egr2 knock-in and CD2-Egr2/3[−/−] mice. **(B)** Analysis of the indicated phenotypic markers in naïve, Egr2[+] MP, Egr2[−] MP, and Egr2/3[−/−] MP CD4 cells from GFP-Egr2 and CD2-Egr2/3[−/−] mice. **(C, D)** Naïve (CD44[lo]CD62L[+]), Egr2[−] MP (GFP-Egr2[−]CD44[hi]CD62L[−]), and Egr2[+] MP CD4 (GFP-Egr2[+]CD44[hi]CD62L[−]) T cells were isolated from GFP-Egr2 knock-in mice (CD45.2) and adoptively transferred into separate wild-type mice (CD45.1). 3 wk after transfer, GFP-Egr2 expression in recipient mice was analysed. **(A, B, C)** are representative of three independent experiments. Data in (D) are the mean ± SD from groups of four recipient mice from one experiment and was analysed with a Kruskal–Wallis test, followed by a Conover test with Benjamini–Hochberg correction. N.S., not significant, *P < 0.05, **P < 0.01.

Interestingly, all CD44high MP CD4 T cells were PD-1high in CD2-Egr2/3−/− mice, whereas the proportions of FoxP3+ Tregs were similar to GFP-Egr2 knock-in mice (Fig 1B), indicating that Egr2/3 are not required for the development of PD-1high MP CD4 T cells, but control their homeostasis and function.

## Egr2/3 are essential for the homeostatic proliferation of PD-1high MP CD4 T cells

To analyse the effect of Egr2/3 on the homeostasis of PD-1high MP CD4 T cells, chimeric mice reconstituted with a mixture of bone marrow from GFP-Egr2 knock-in (CD45.1) and CD2-Egr2/3−/− (CD45.2) mice were established allowing the development of Egr2+ and Egr2/3−/− PD-1high MP cells in the same environment. Naïve, Egr2+PD-1high MP (Egr2+CD44highPD-1high), Egr2− MP (Egr2−CD44highPD-1low), and Egr2/3−/− PD-1high MP (Egr2/3−/−CD44highPD-1high) CD4 T cells from chimeras were analysed for Ki67, a proliferation marker. Similar, low percentages of Ki67-positive cells were found among naïve T cells of GFP-Egr2 knock-in and CD2-Egr2/3−/− origin (Fig 2A). Nearly half of Egr2+PD-1high MP cells were Ki67+, whereas the percentages of Ki67+ cells were lower in Egr2−PD-1low MP and much lower in the Egr2/3−/− PD-1high MP populations (Fig 2A and B). To analyse homeostatic proliferation, Egr2+PD-1high MP and Egr2−PD-1low MP CD4 T cells of GFP-Egr2 knock-in (CD45.1) origin, and Egr2/3−/− PD-1high MP CD4 cells of CD2-Egr2/3−/− (CD45.2) origin

were isolated from the chimeras. Cells were labelled with Cell-Trace Violet before adoptive transfer into wild-type (CD45.1/2) mice. Egr2+PD-1high MP cells were highly proliferative with more than 75% of cells having divided at least once, whereas more than half of Egr2−PD-1low MP cells had not divided and those that had underwent fewer divisions than Egr2+PD-1high MP cells (Fig 2C). In contrast, Egr2/3−/− PD-1high MP cells hardly proliferated, with more than three-quarters of cells not undergoing any homeostatic proliferation in recipient mice (Fig 2C). The results demonstrate that Egr2/3 support the homeostatic proliferation of PD-1high MP cells in the steady state.

## Egr2/3 regulate genes required to maintain the homeostasis of PD-1high MP CD4 T cells

To understand the mechanisms of Egr2 and Egr3 function in PD-1high MP T cells, we analysed the target genes of Egr2 in CD4 T cells and the transcriptomes of Egr2+PD-1high MP, Egr2−PD-1low MP, and Egr2/3−/−PD-1high MP CD4 cells from GFP-Egr2 knock-in and CD2-Egr2/3−/− mice at 7 wk of age. At this time point, T cells from CD2-Egr2/3−/− mice have not developed into pathogenic cells and do not express inflammatory cytokines, such as IFN gamma (IFNγ, or activation markers, such as CD69 (Li et al, 2012).

We focussed on comparing GFP-Egr2+ MP to GFP-Egr2− MP and GFP-Egr2+ MP to Egr2/3−/− MP T cells (Tables S1 and S2). Of those

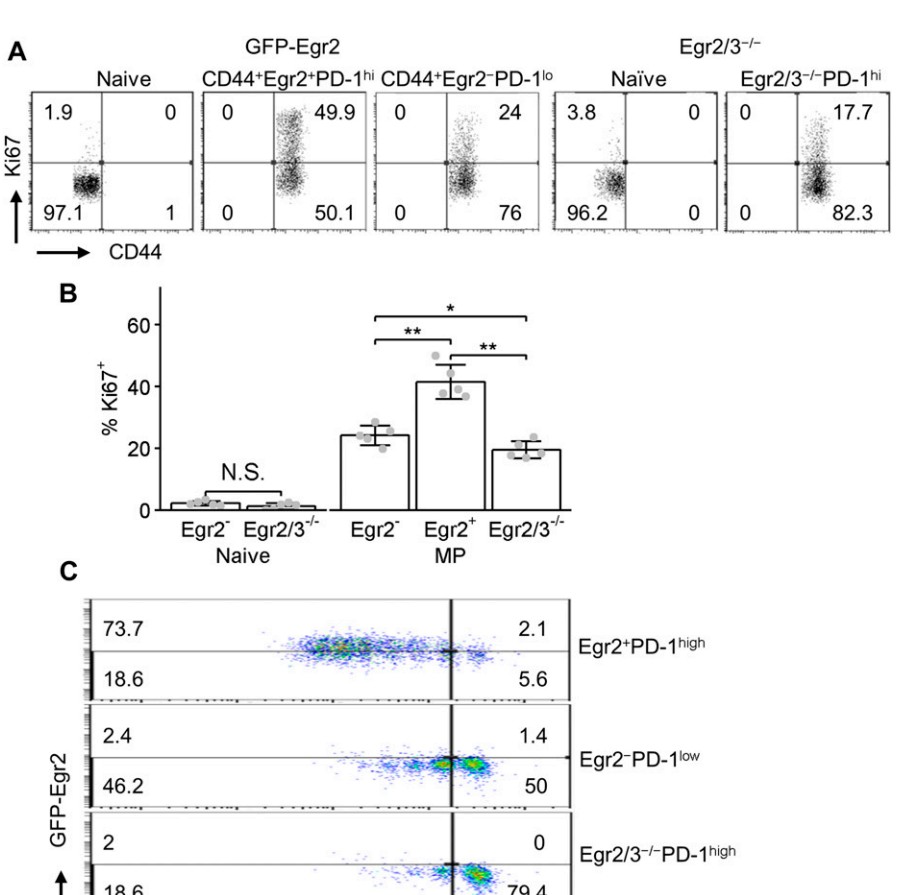

**Figure 2. Egr2 and 3 maintain the homeostatic proliferation of PD-1high memory phenotype (MP) CD4 T cells.**
Chimeric mice were generated by reconstitution with mixed BM from GFP-Egr2 knock-in (CD45.1) and CD2-Egr2/3−/− (CD45.2) mice. **(A, B)** Ki67 and CD44 expression in gated naïve, GFP-Egr2− MP, GFP-Egr2+PD-1high MP, and Egr2/3−/−PD-1high MP CD4 T cells from spleens and lymph nodes of chimeric mice 8–12 wk after reconstitution. **(C)** MP CD4 T cells (CD62L−CD44hi) of GFP-Egr2 knock-in (CD45.1) and CD2-Egr2/3−/− (CD45.2) origin were isolated from chimeric mice and mixed in equal numbers before labelling with CellTrace Violet. The labelled cells were adoptively transferred into wild-type recipients (CD45.1/2). 3 wk after transfer, CellTrace Violet was analysed on gated GFP-Egr2+ PD-1high, GFP-Egr2−PD-1low (both CD45.1), and Egr2/3−/− PD-1high (CD45.2) donor cells. **(A, C)** are representative of three independent experiments. Data in (B) are the mean ± SD from groups of five recipient mice from one experiment and were analysed with Kruskal–Wallis tests, followed by Conover tests with Benjamini–Hochberg correction. N.S., not significant, *P < 0.05, **P < 0.01.

genes that were differentially expressed in either comparison, around a third were changed in both comparisons. Differentially expressed genes associated with T-cell biology included those involved in DNA repair (e.g., Ung, Mgmt, and Apex1), cell survival and growth (such as Myb, Rel, Eomes, Gfi1, Bcl6, and Id3), metabolism (such as Scd1, Scd2, and solute channels) and homeostasis (such as P2rx7, Il2, and Il2ra), which were up-regulated in GFP-Egr2+ PD-1high MP cells, and also those involved in inflammatory responses (such as Runx2, Tbx21, Ahr, Rorc, Il18r1, Il18rap, Icam1, Il23r, Il17re, Il12rb2, Csf2rb2, and chemokines or chemokine receptors), which were down-regulated (Fig 3A). A gene set enrichment analysis type approach indicated that pathways involved in proliferation and metabolism, such as Myc targets, Ras signalling, and Heme metabolism, were increased in GFP-Egr2+ MP T cells, compared with

either GFP-Egr2− MP or Egr2/3−/− MP T cells, whereas pathways involved in inflammation, such as allograft rejection and IFN response, were reduced (Fig 3B). Compared with GFP-Egr2− cells, Egr2/3−/− MP cells also had reduced expression of additional genes involved in cell growth and homeostasis (such as E2f1, Cdk1, Runx1, Tgfb1, and Lif) and increased expression of further inflammatory genes (such as Il21, Il1r2, and additional chemokine receptors), indicating more profound homeostatic defects. Overall, these results show that despite their common expression of high levels of PD-1, Egr2/3−/− MP and GFP-Egr2+ MP T cells are distinct, with Egr2+ MP T cells being more proliferative and less inflammatory than either Egr2− or Egr2/3−/− MP cells. In summary, Egr2 and Egr3 regulate the expression of genes involved in proliferation and metabolism while suppressing the expression of inflammatory pathways in PD-1high MP CD4 T cells.

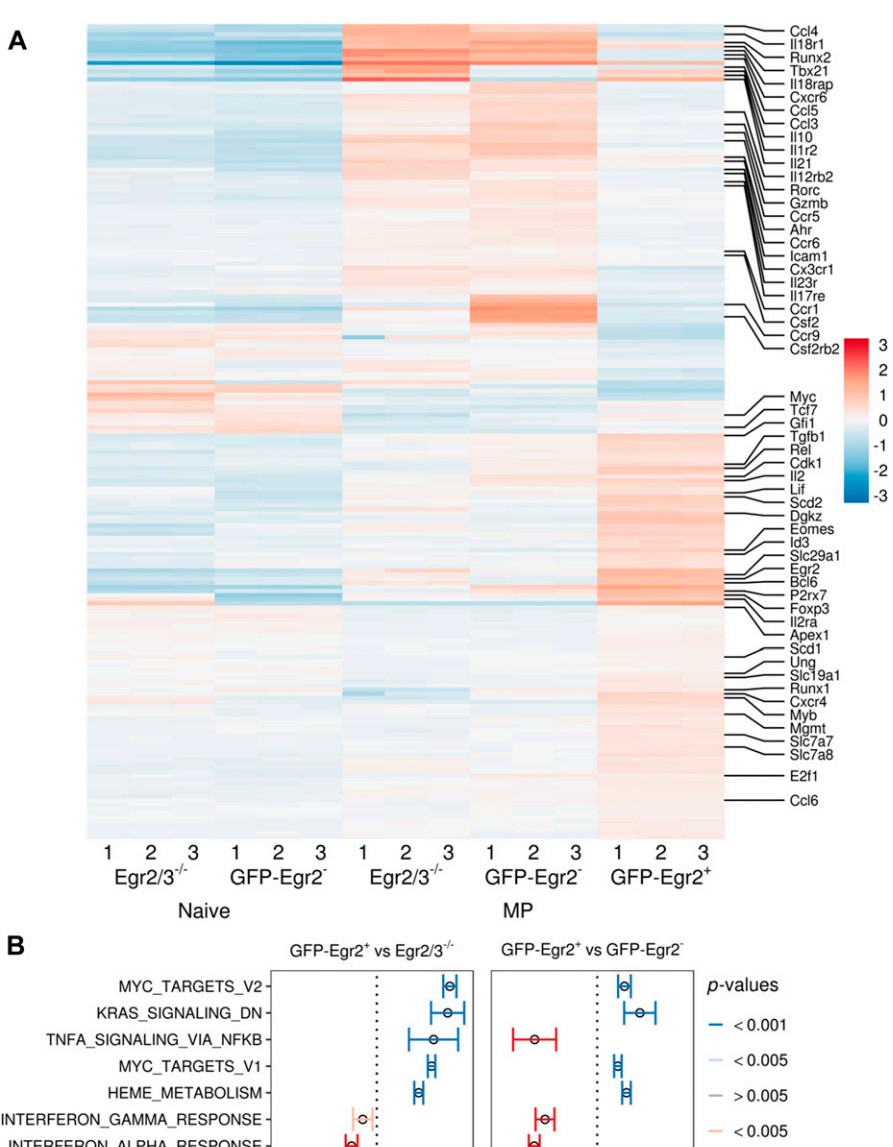

**Figure 3. Egr2 and 3 reciprocally regulate homeostatic and inflammatory programmes in memory phenotype (MP) T cells.**
Naïve (CD62L+CD44lo) and GFP-Egr2− MP and GFP-Egr2+ MP from GFP-Egr2 knock-in and naïve and MP Egr2/3−/− from CD2-Egr2/3−/− mice were analysed by RNA-seq. **(A)** Unsupervised hierarchical clustering of selected genes showing expression patterns in naïve, GFP-Egr2−, GFP-Egr2+, and Egr2/3−/− MP T cells. Selected genes relevant to MP T cell function are indicated. **(B)** Gene set enrichment analysis of Hallmark gene sets (Liberzon et al, 2015) for GFP-Egr2+ versus Egr2/3−/− MP cells (left) and GFP-Egr2+ versus GFP-Egr2− MP cells (right). Mean and 95% confidence intervals for selected pathways, colour coded to indicate Benjamini–Hochberg corrected P-values, are shown. The RNA-seq data are from three biological replicates, each with cells pooled from 10 mice, for each group.

To investigate the target genes of Egr2, naïve CD4 T cells were isolated from GFP-Egr2 knock-in mice and stimulated in vitro with anti-CD3 and anti-CD28 to induce GFP-Egr2 expression before GFP-Egr2-chromatin immunoprecipitation-sequencing (ChIP-seq) analysis. We found that the anti-Egr2 antibody used for ChIP-seq in previous reports (Zheng et al, 2013; Du et al, 2014) was highly cross-reactive, and we could not get consistent results from replicated experiments in CD4 T cells (Fig S1). Taking advantage of our GFP-Egr2 knock-in model, we used the GFP-Trap ChIP method which produced high quality Egr2 ChIP from repeated experiments as indicated by enrichment of known target genes such as *Nab2* and *Tcf7* (Fig S1). To define high confidence peaks, we used the irreproducible discovery rate (IDR) method (Landt et al, 2012) to identify peaks detected in replicates. The enriched motif identified in these peaks (Fig 4A) was highly consistent with the Egr2-binding consensus sequence defined previously (Sham et al, 1993; Swirnoff & Milbrandt, 1995). Most of the validated Egr2 target genes in previous reports, such as *Fasl*, *Nab2*, *Dgkz*, *Tcf7*, and *Bcl6*, were among the target genes detected (Rengarajan et al, 2000; Zheng et al, 2013; Du et al, 2014; Ogbe et al, 2015; Miao et al, 2017). Egr2 predominantly interacted with gene promoters (Fig 4B), and target genes associated with T-cell function were mostly involved in proliferation, metabolism, and regulation of inflammation (Fig 4C). About a quarter of genes that were differentially expressed between GFP-Egr2+ PD-1^{high} MP CD4 T cells and either GFP-Egr2− or Egr2/3^{−/−} MP CD4 T cells were target genes of Egr2 (Fig 4D). Many of the genes that are functionally associated with proliferation defects and high inflammation of Egr2/3^{−/−} PD-1^{high} MP CD4 T cells such as *P2rx7*, *Myc*, *Il2ra* (down-regulated in Egr2/3^{−/−} MP cells), and *Icam1* (up-regulated in Egr2/3^{−/−} MP cells) are Egr2 targets (Fig 4E and F).

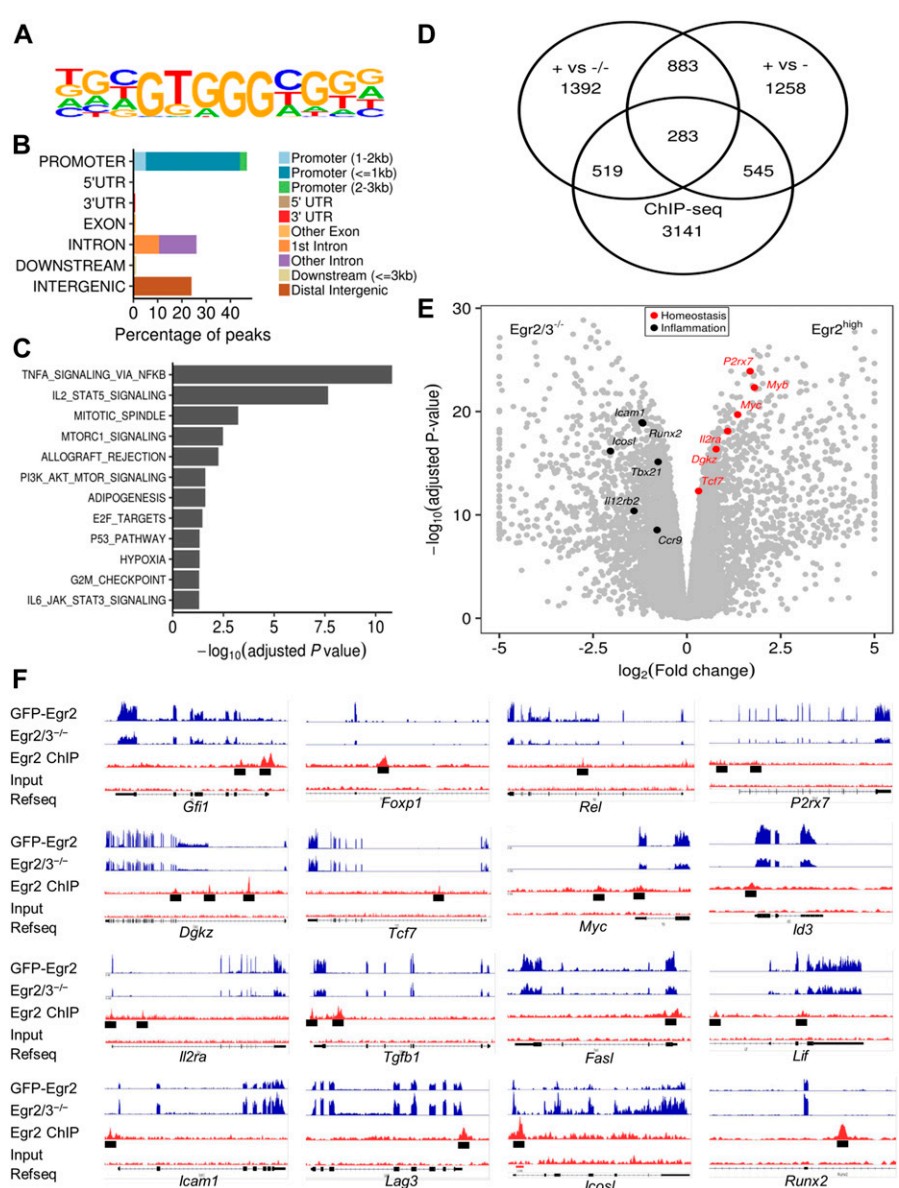

**Figure 4. Egr2 binds to regulatory regions of genes involved in homeostasis and control of inflammatory responses of Egr2+ memory phenotype (MP) T cells.**
CD4 T cells from GFP-Egr2 knock-in mice were stimulated for 24 h in vitro with anti-CD3 and anti-CD28 to induce GFP-Egr2 expression and then used for GFP-Egr2-ChIP-seq. **(A)** Most significant motif enriched in Egr2 ChIP-seq peaks (*P* = 1 × 10^1465). **(B)** Distribution of Egr2 binding sites in the genome. **(C)** Functional analysis of the genes bound in Egr2 ChIP-seq using the Hallmark gene sets (Liberzon et al, 2015). **(D)** Proportion of differentially expressed genes in RNA-seq (Fig 3) that are bound by Egr2; "+ versus −/−" and "+ versus −" indicate the GFP-Egr2+ MP versus Egr2/3^{−/−} MP and GFP-Egr2+ MP versus GFP-Egr2− MP comparisons in RNA-seq, respectively. **(E)** Volcano plot of RNA-seq data for GFP-Egr2+ MP versus Egr2/3^{−/−} MP cells, with positive and negative log₂ fold changes indicating higher expression in GFP-Egr2+ or Egr2/3^{−/−} cells, respectively. Selected genes bound in GFP-Egr2-ChIP-seq are indicated. **(F)** ChIP-seq peaks (third track for each gene) associated with the indicated genes, together with RNA-seq reads from GFP-Egr2+ and Egr2/3^{−/−} MP cells (top two tracks), compared with signal from Input chromatin (fourth track). The ChIP-seq data are from three independent IPs each from an independent biological replicate.

These data show that Egr2/3 are essential for regulating genes involved in proliferation and homeostasis of PD-1[high] MP T cells.

## Egr2/3 regulate the fitness of PD-1[high] MP CD4 T cells for adaptive immune responses

To assess Egr2/3 function in regulation of adaptive responses of MP CD4 T cells, we used an OT-II retrogenic model (Holst et al, 2006b; Miao et al, 2017) to generate MP T cells that have not encountered antigen in the steady state. Before reconstitution, a mixture of bone marrow from wild-type (CD45.1), and CD2-Egr2/3[−/−] (CD45.2) mice was transduced with retrovirus carrying OT-II TCR genes and a GFP reporter gene as described (Miao et al, 2017) (Fig S2A). The OT-II expressing T cells were analysed by I-A[b]-OVA[329-337] tetramer (Fig S2B). 8 wk after reconstitution, OT-II cells of both genotypes were detected in chimeric mice (Fig S2B). A proportion of OT-II T cells of both wild-type and Egr2/3[−/−] origin developed into CD44[high] MP cells in the steady state (Fig 5A and B). To assess the homeostasis of MP OT-II T cells of wild-type and Egr2/3[−/−] origin, MP CD4 T cells were isolated and equal numbers of wild-type (CD45.1) and Egr2/3[−/−] (CD45.2) cells were combined, before adoptive transfer into wild-type recipients (CD45.1/2). 24 h after transfer, the numbers of wild-type and Egr2/3[−/−] MP OT-II (Fig 5C and D) cells were similar. However, 3 wk after transfer, the numbers of Egr2/3[−/−] MP cells were significantly reduced compared to wild-type counterparts (Fig 5C and D). The expression of Egr2 regulated genes (*Myb*, *Tcf7*, *P2rx7*, and *Icam1*) by MP OT-II cells 3 wk after transfer showed reduced

expression of Myb, Tcf7, and P2rx7 and increased expression of Icam1 in Egr2/3[−/−] MP cells compared with wild-type counterparts (Fig 5E). These results indicate that the homeostatic maintenance of antigen-inexperienced PD-1[high] MP T cells is regulated by Egr2/3.

To assess the function of Egr2/3 in adaptive immune responses of antigen-inexperienced PD-1[high] MP T cells, a mixture of wild-type and Egr2/3[−/−] PD-1[high] MP OT-II cells was transferred as above and recipient mice were infected with OVA-vaccinia virus 24 h later as described in our report (Miao et al, 2017). 7 d after infection, wild-type donor cells had expanded, whereas the expansion of Egr2/3[−/−] donor cells was impaired (Fig 5F and G). Consistent with this, Ki67 positive cells were significantly reduced in Egr2/3[−/−] donor cells compared with wild-type counterparts (Fig 5H and I). IFNγ was produced by a proportion of wild-type cells consistent with previous findings (Román et al, 2002; Foulds & Shen, 2006; Miao et al, 2017). Interestingly, although Egr2/3[−/−] OT-II PD-1[high] MP cells failed to expand in response to viral infection, more of them produced IFNγ than their wild-type counterparts (Fig 5J and K). These results demonstrate the importance of Egr2/3 in adaptive responses of PD-1[high] MP T cells to pathogens.

## Reduced repertoire diversity of MP CD4 T cells from CD2-Egr2/3[−/−] mice

MP T cells in the periphery are largely maintained by homeostatic cytokines, such as IL-7 (Boyman et al, 2007). We found that CD127 levels were similar between GFP-Egr2[+] and Egr2/3[−/−] MP T cells (Fig

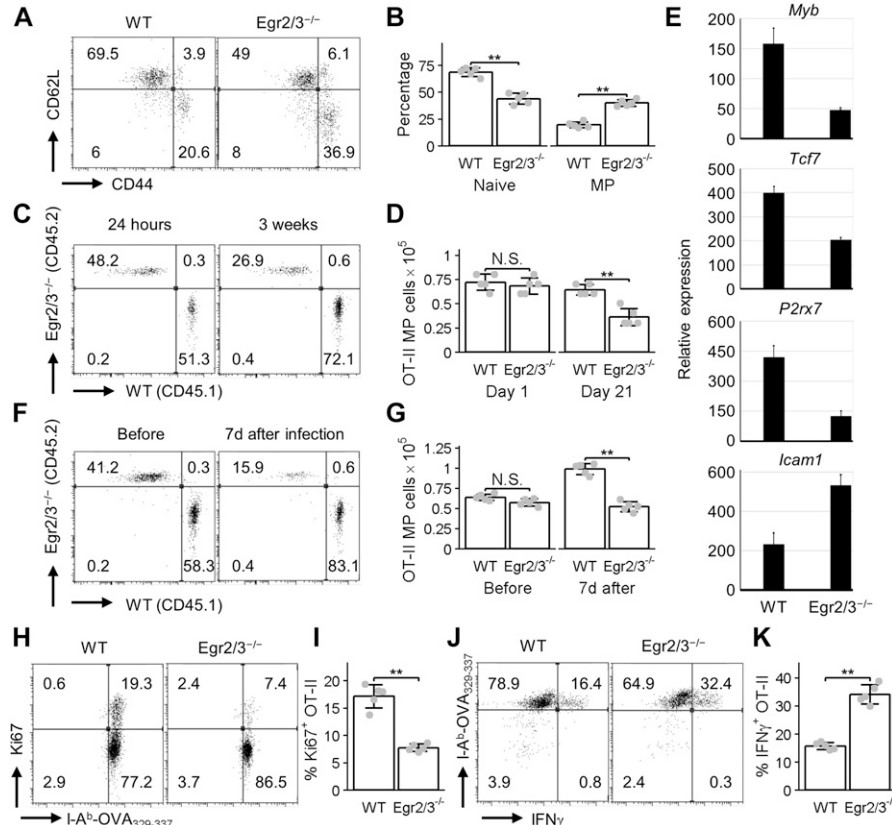

**Figure 5. Antigen-inexperienced memory phenotype (MP) T cells are intrinsically regulated by Egr2 and 3 for efficient adaptive immune responses.**
Mixed BM OT-II TCR retrogenic chimera models were created by adoptive transfer of an equal number of OT-II retrovirus transduced BM cells from wild-type (CD45.1) and CD2-Egr2/3[−/−] (CD45.2) mice. (A, B) 8 wk after BM reconstitution, OT-II (I-A[b]-OVA[+][329-337]CD4[+]) wild-type (CD45.1) and Egr2/3[−/−] (CD45.2) cells from spleens and lymph nodes of chimeras were analysed for expression of CD62L and CD44. (C, D, E, F, G, H, I, J, K) GFP[+]CD62L[−]CD44[hi] MP OT-II cells were isolated from the chimeras and equal numbers of wild-type (CD45.1) and Egr2/3[−/−] (CD45.2) MP OT-II cells were adoptively transferred into wild-type mice (CD45.1/2). (C, D) The percentages (C) and absolute numbers (D) of donor cells of each genotype were assessed 24 h or 3 wk after transfer. (E) RT-PCR of the indicated genes in isolated OT-II wild-type or Egr2/3[−/−] donor cells 3 wk after transfer. (F, G, H, I, J, K) 7 d after transfer, a group of recipient mice were infected with OVA-vaccinia virus i.p. and the percentage (F) and absolute number (G) of wild-type and Egr2/3[−/−] donor cells were analysed before and 7 d after infection. (H, I, J, K) 7 d after infection, Ki67-positive (H, I) and IFNγ-producing (J, K) OT-II cells were analysed. (A) is representative of 15 recipient mice. (C, E, F, H, J) are representative of two to three experiments with similar results. Data in (B, D, G, I, K) are the mean ± SD of five recipient mice and were analysed with Mann–Whitney two-tailed tests. N.S., not significant, *P < 0.05, **P < 0.01.

6A and B), indicating that the impaired homeostatic proliferation of Egr2/3$^{-/-}$ MP T cells is not due to lack of IL-7Rα. We also examined expression of CD5, the levels of which are associated with TCR affinity for self-peptide MHC complexes (Kawabe et al, 2017). CD5 expression was similar in naïve T cells from GFP-Egr2 knock-in and CD2-Egr2/3$^{-/-}$ mice (Fig 6A and B). Around a quarter of Egr2$^-$ MP T cells were CD5$^{high}$, whereas this was increased to around a third in the Egr2$^+$ MP population (Fig 6A and B). In contrast, more than half of Egr2/3$^{-/-}$ MP T cells expressed high levels of CD5 (Fig 6A and B), indicating that these cells may be auto-reactive.

Homeostatic proliferation maintains the repertoire diversity of T cells which is important for sustaining adaptive immunity especially after thymic involution (Qi et al, 2014; Lanzer et al, 2018). To assess whether the altered homeostatic proliferation of Egr2/3$^{-/-}$ MP cells CD4 T cells changes their repertoire diversity, we analysed the TCR repertoires of total MP (CD25$^-$CD62L$^-$CD44$^{hi}$) CD4 T cells from wild-type and CD2-Egr2/3$^{-/-}$ mice. TCRVβ, TCRJβ, and CDR3-encoding junctional sequences were compared from three independent replicate experiments. Naïve TCRβ repertoire diversity was similar between wild-type and Egr2/3$^{-/-}$ naïve CD4 T cells (Fig 6C). MP CD4 T cells from wild-type mice had reduced TCRβ repertoire diversity compared with naïve counterparts (Fig 6C), which is consistent with previous reports (Qi et al, 2014). However, the repertoire diversity of Egr2/3$^{-/-}$ MP CD4 cells was profoundly reduced compared with wild-type MP T cells (Fig 6C),

indicating that the impaired homeostatic proliferation of Egr2/3$^{-/-}$ MP CD4 altered their diversity. Analysis of clonal frequency plotted against clonal rank showed a significant enrichment of a few clones in Egr2/3$^{-/-}$ MP T cells (Fig 6D). The increased proportion of CD5$^{high}$ cells among Egr2/3$^{-/-}$ MP CD4 T cells (Fig 6A and B), and auto-reactive T cells and autoimmune disease in CD2-Egr2/3$^{-/-}$ mice (Li et al, 2012; Morita et al, 2016) suggests that these enriched clones may have high affinity for self-antigen.

### Egr2/3 control inflammatory responses of PD-1$^{high}$ MP T cells

Despite high PD-1 expression, Egr2/3$^{-/-}$ MP CD4 T cells are highly inflammatory leading to the development of autoimmune disease (Li et al, 2012; Morita et al, 2016). Egr2/3 are only expressed in PD-1$^{high}$ MP CD4 T cells (Fig 1A and B). To investigate whether Egr2/3 control the inflammatory responses of PD-1$^{high}$ MP T cells to inflammatory cytokine stimulation, Egr2$^+$PD-1$^{high}$ MP, Egr2$^-$PD-1$^{low}$ MP, and Egr2/3$^{-/-}$ PD-1$^{high}$ MP CD4 T cells from GFP-Egr2 knock-in and CD2-Egr2/3$^{-/-}$ mice were stimulated in vitro with IL-12. Very few Egr2$^+$PD-1$^{high}$ MP cells produced IFNγ in response to IL-12 stimulation, whereas IL-12 elicited IFNγ production by a small proportion of Egr2$^-$PD-1$^{low}$ MP cells (Fig 7A and B). However, IFNγ producing Egr2/3$^{-/-}$ PD-1$^{high}$ MP CD4 T cells were significantly increased in response to IL-12 (Fig 7A and B). T-bet has been reported to play an

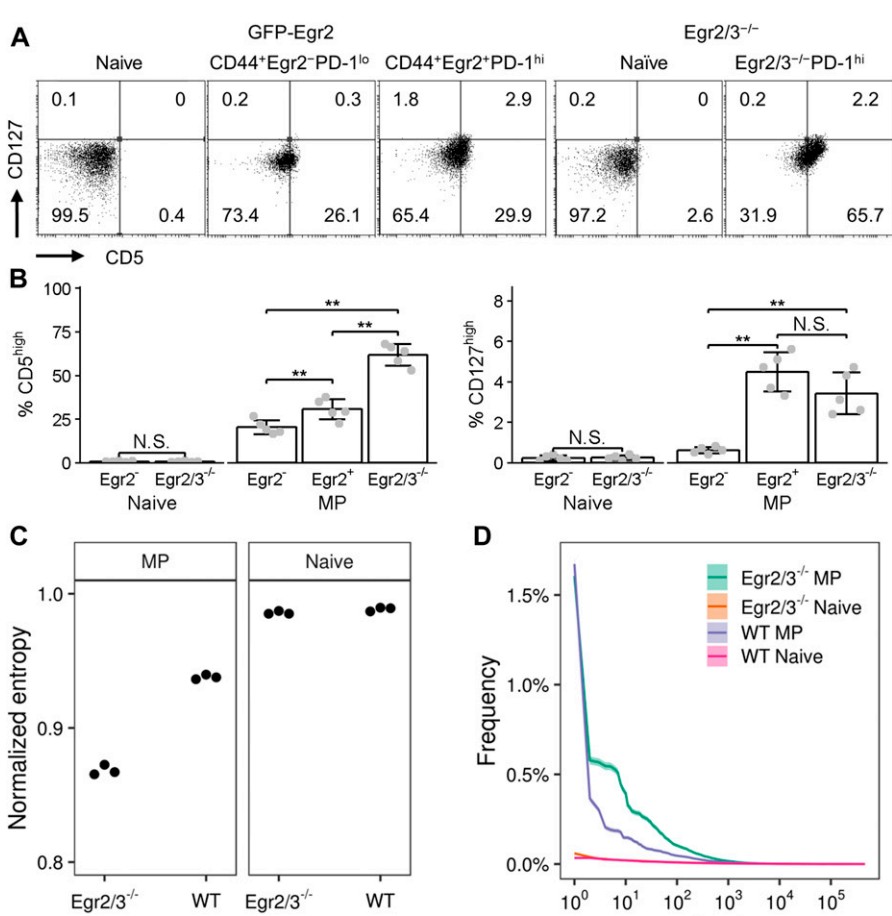

**Figure 6. Reduced repertoire diversity of CD2-Egr2/3$^{-/-}$ memory phenotype (MP) T cells.**
**(A, B)** Naïve and GFP-Egr2$^+$, GFP-Egr2$^-$, and Egr2/3$^{-/-}$ MP T cells from GFP-Egr2 knock-in and CD2-Egr2/3$^{-/-}$ mice were analysed for CD127 and CD5 expression. **(C, D)** CD4 naïve and MP T cells were isolated from wild-type (WT) and CD2-Egr2/3$^{-/-}$ mice and their TCRβ repertoires analysed. **(C)** Repertoire diversity was estimated using the Shannon entropy index normalized by the total number of unique amino acid clonotypes. Samples were downsampled to the size of the smallest repertoire 100 times and the Shannon entropy index calculated for each. The median of the 100 diversity estimates for each sample is plotted. **(D)** Rank frequency distribution of MP and naïve T cell clonotypes from wild-type and CD2-Egr2/3$^{-/-}$ mice. Clonotype frequency was estimated using the three replicates for each condition using the Chao1 estimator. Clonotype rank against frequency in the repertoire is shown. The TCR-seq data are from three biological replicates, each with cells pooled from 10 mice, for each group. Data in (A) are representative of three independent experiments. Data in (B) are the mean ± SD from groups of mice and were analysed with Kruskal–Wallis tests, followed by Conover tests with Benjamini–Hochberg correction. N.S., not significant, *$P < 0.05$, **$P < 0.01$.

important role in innate-like inflammatory responses of CD4 PD-1[high] MP T cells (Kawabe et al, 2017) and Egr2/3 are repressors of T-bet function (Singh et al, 2017). The percentage of T-bet[+] MP CD4 T cells was higher in Egr2[−]PD-1[low] MP cells than Egr2[+]PD-1[high] MP T cells from GFP-Egr2 knock-in mice (Fig 7C). However, T-bet expression was significantly increased in Egr2/3[−/−] PD-1[high] MP CD4 T cells (Fig 7C). Thus, increased T-bet levels and/or activity in the absence of Egr2/3 may play a key role in inflammatory responses of Egr2/3[−/−] PD-1[high] MP T cells and in the development of inflammatory autoimmune diseases.

## Defective expression of Egr2 in PD-1[high] MP CD4 T cells from RA patients

It has recently been found that PD-1[high] MP CD4 T cells accumulate in joint synovial tissue and in the peripheral blood of patients with active RA and SLE (Rao et al, 2017; Bocharnikov et al, 2019; Caielli et al, 2019; Zhang et al, 2019). The phenotype, inflammatory activation, and cytokine profile of these PD-1[high] MP CD4 T cells from synovial

tissues resembles PD-1[high] MP CD4 T cells from CD2-Egr2/3[−/−] mice (Figs 1B and 3A). Therefore, we assessed the expression of Egr2 in PD-1[high] MP CD4 T cells from the peripheral blood of patients with active RA (Table S3). PD-1[high] MP CD4 T cells (PD-1[high]CD45RA[−]) were detected in both healthy controls and patients (Fig 8A) and most PD-1[high] MP CD4 T cells were CXCR3[+]CXCR5[−] (Fig S3A). PD-1[high] MP CD4 T cells were increased in patients compared with healthy controls (Fig 8A and B), consistent with previous findings (Rao et al, 2017; Zhang et al, 2019). Egr2 was expressed in a proportion of PD-1[high] MP T cells in healthy controls but was significantly reduced in patients with active RA (Fig 8A and B). Taken together with our findings from mice, this suggests that Egr2 and/or Egr3 are intrinsic regulators of PD-1[high] MP CD4 T cells to maintain their homeostasis and to prevent autoimmune inflammation in the steady state.

We previously found that Egr2/3 can suppress the activity of the Th1 transcription factor T-bet (Singh et al, 2017). Granzyme B and T-bet expression were increased in Egr2/3[−/−] in PD-1[high] MP CD4 T cells (Figs 3A and 7C and D). These two molecules are highly expressed in PD-1[high] MP CD4 T cells from joint synovial tissue of

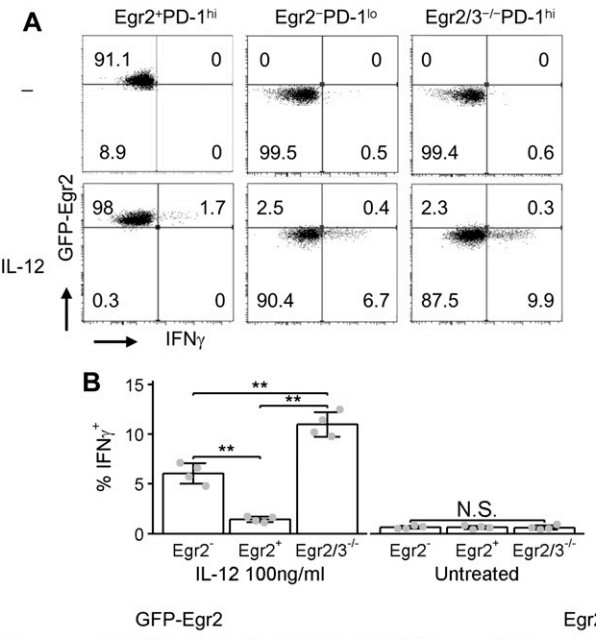

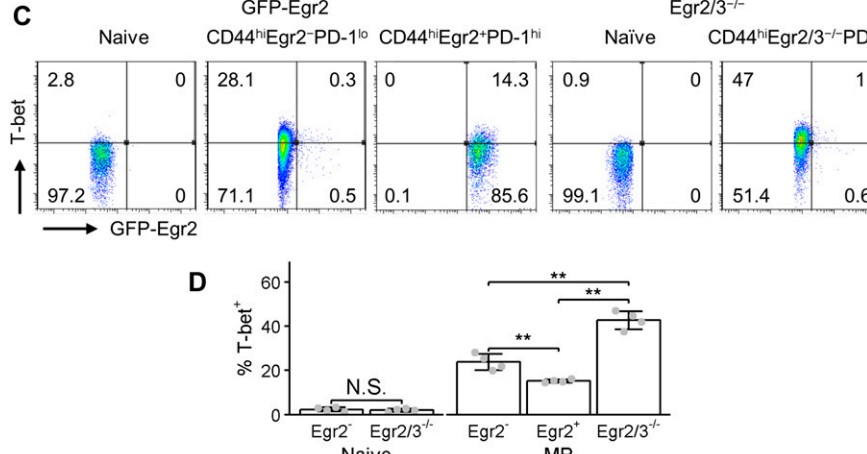

**Figure 7. Egr2 and 3 control IFNγ production by CD4 memory phenotype (MP) T cells in response to IL-12 stimulation.**

**(A, B)** GFP-Egr2[+]CD4[+]CD25[−]CD62L[−]CD44[hi] and GFP-Egr2[−]CD4[+]CD25[−]CD62L[−]CD44[hi] MP T cells were isolated from GFP-Egr2 knock-in and CD4[+]CD25[−]CD62L[−]CD44[hi] MP T cells were isolated from CD2-Egr2/3[−/−] mice and stimulated in vitro with 100 ng/ml IL-12 for 24 h before analysis of GFP-Egr2 and IFNγ by flow cytometry.
**(C, D)** Naïve and GFP-Egr2[+], GFP-Egr2[−] and Egr2/3[−/−] MP T cells from GFP-Egr2 knock-in and CD2-Egr2/3[−/−] mice were analysed for GFP-Egr2 and T-bet expression. Data are representative of three to four experiments. Data in (B, D) are the mean ± SD of four samples and were analysed with Kruskal–Wallis tests, followed by Conover tests with Benjamini–Hochberg correction. N.S., not significant, *P < 0.05, **P < 0.01.

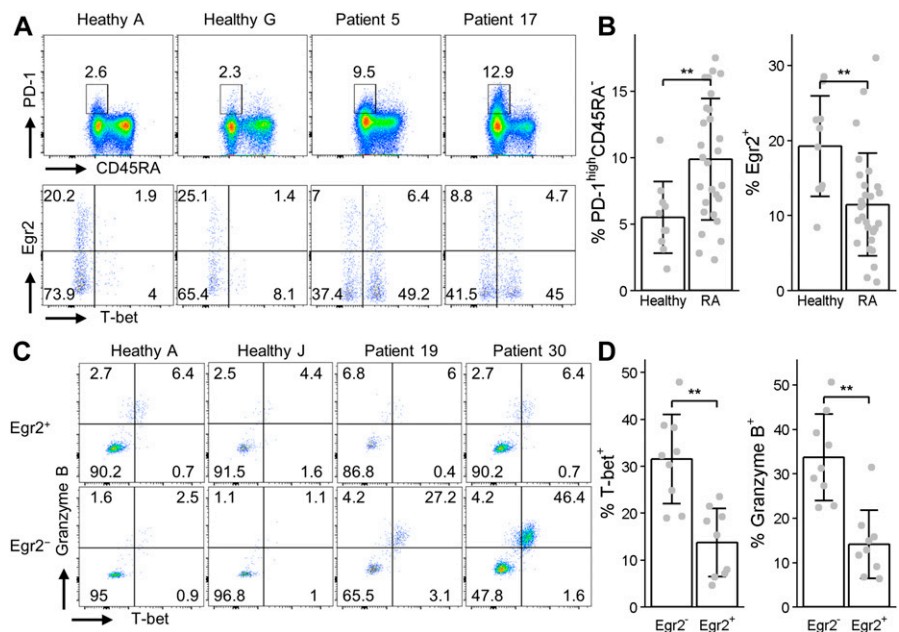

**Figure 8.   Egr2 expression is reduced in PD-1^high CD4 memory phenotype T cells from patients with active rheumatoid arthritis.**
**(A, B)** CD3+CD4+ cells from PBMCs of healthy controls and rheumatoid arthritis patients were analysed for PD-1 and CD45RA expression by flow cytometry (A top panel) and the proportion of PD-1^high CD45RA− cells was quantified (B, left panel). Egr2 and T-bet expression by these gated PD-1^high CD45RA− cells was then analysed by flow cytometry (A, bottom panel) and the proportion of Egr2+ cells was quantified (B, right panel). **(C, D)** Patients in which more than 10% of PD-1^high CD45RA− cells were T-bet positive were gated on Egr2− and Egr2+ cells and analysed for T-bet and Granzyme B expression. Healthy controls in (C) are presented for comparison. Data in (B, D) are the mean ± SD and were analysed with Mann–Whitney two-tailed tests. N.S., not significant, *P < 0.05, **P < 0.01.

arthritis patients (Rao et al, 2017). Although overall there was not a significant difference between patients and controls in the proportions of PD-1^high MP CD4 T cells expressing T-bet and Granzyme B (Fig S3B), in patients in which more than 10% of PD-1^high MP CD4 T cells expressed T-bet, we found that more T-bet+ and Granzyme B+ cells were detected in Egr2− PD-1^high MP CD4 T cells than in Egr2+ PD-1^high MP CD4 T cells (Fig 8C and D). This may require further investigation to assess the pathological impact of these cells. The reduced expression of Egr2 coupled with increased T-bet and Granzyme B expression in PD-1^high CD4 T cells may serve as one of the molecular signatures for active arthritis.

# Discussion

Checkpoint molecules are expressed in effector or effector phenotype T cells in both acute and chronic infections which is important to limit immunopathology and can also lead to exhaustion (Wherry, 2011; Zhang & Vignali, 2016). Recently, it has been shown that a subset of MP CD4 T cells express high levels of PD-1 (PD-1^high MP) and have pathological function in the development of RA and SLE (Rao et al, 2017; Bocharnikov et al, 2019; Caielli et al, 2019; Zhang et al, 2019). We have now shown that Egr2 is highly expressed in PD-1^high MP CD4 T cells in both mice and humans in the steady state. Egr2 plays an essential role to support homeostatic proliferation and control the inflammatory function of these cells by regulating genes in Myc, mTORC, IL-2 signalling, and metabolic pathways as well as genes linked to allograft rejection and IFN-mediated inflammation in a reciprocal fashion. In the absence of Egr2/3, PD-1^high MP CD4 T cells are highly inflammatory but have impaired homeostasis and T-cell function, a phenotype discovered recently in SLE and arthritis (Rao et al, 2017; Tilstra et al, 2018; Arazi et al,

2019). Together with the defective expression of Egr2 in PD-1^high MP CD4 T cells from peripheral blood of patients with active RA, our findings indicate that Egr2 and/or 3 are essential regulators for the control of inflammatory function and homeostatic fitness of PD-1^high MP CD4 T cells.

PD-1 controls the proliferation and autoimmune responses of CD4 T cells (Okazaki et al, 2013). The checkpoint molecule Lag3 also has an inhibitory function in T cells (Okamura et al, 2009). They both can be induced in effector T cells in acute viral infection as well as in chronic infection and cancer. We and others found that PD-1 is highly expressed in a subset of MP CD4 (PD-1^high MP) T cells under steady state conditions (Fig 1B) (Rao et al, 2017; Bocharnikov et al, 2019; Caielli et al, 2019; Zhang et al, 2019). In SLE and RA, PD-1^high MP CD4 T cells accumulate and are inflammatory (Rao et al, 2017; Bocharnikov et al, 2019; Caielli et al, 2019; Zhang et al, 2019). We have now demonstrated that Egr2 is highly expressed in PD-1^high MP CD4 T cells in the steady state. In addition to high levels of PD-1, Egr2+ and Egr2/3^−/− MP T cells express many of the same markers, such as CXCR3, that have been described in pathogenic PD-1^high MP T cells in disease (Rao et al, 2017; Bocharnikov et al, 2019; Caielli et al, 2019). We have now discovered that the homeostasis and inflammatory activity of PD-1^high MP CD4 T cells is regulated by Egr2/3 in the steady state. Egr2/3 are not required for the development of the PD-1^high MP CD4 T cell population but support their homeostatic proliferation and control their inflammatory function.

We and others previously showed that Egr2/3 deficiency results in accumulation of hyper-activated inflammatory MP CD4 T cells leading to severe autoimmune responses (Li et al, 2012; Morita et al, 2016). We have also previously observed that Egr2^−/− MP CD4 T cells in old CD2- Egr2^−/− mice, which are also prone to develop autoimmunity (Zhu et al, 2008; Miao et al, 2013), have increased homeostatic proliferation (Zhu et al, 2008). In contrast, we have now demonstrated that the homeostatic proliferation of Egr2/3^−/− MP

CD4 T cells from healthy chimeric mice is impaired (Fig 2), indicating that Egr2/3$^{-/-}$ MP CD4 T cells have an intrinsic homeostatic defect. This altered homeostasis of Egr2/3$^{-/-}$ PD-1$^{high}$ MP CD4 T cells results in a skewed MP T cell repertoire with reduced diversity and oligoclonal expansion of MP CD4 T cells with high affinity for self-antigens as indicated by high levels of CD5. Collectively, these data demonstrate that the maintenance of homeostasis is not only important to preserve a diverse T-cell repertoire but also for controlling the expansion of auto-reactive MP CD4 T cells.

Although Egr2$^+$ and Egr2/3$^{-/-}$ PD-1$^{high}$ MP CD4 T cells have a similar cell surface phenotype, Egr2/3$^{-/-}$ PD-1$^{high}$ MP CD4 T cells have an altered expression profile with increased Il21, Il10, Tbx21, Gzmb, and Cx3cr1 and decreased Myc, P2rx7, Il2, and Bcl6. This expression profile partially resembles that found in PD-1$^{high}$ MP CD4 T cells from joint synovial tissue of RA patients (Rao et al, 2017; Zhang et al, 2019) and includes expression of Il21 and Il10 which are important for extrafollicular B-cell helper function by PD-1$^{high}$ MP T cells in autoimmunity (Bocharnikov et al, 2019; Caielli et al, 2019), indicating the importance of Egr2 and/or 3 in the control of the inflammatory function of PD-1$^{high}$ MP CD4 T cells. We have now demonstrated that Egr2 expression is impaired in PD-1$^{high}$ MP CD4 T cells from patients with active RA compared with healthy controls. Although the mechanisms responsible for the down-regulation of Egr2 expression in PD-1$^{high}$ MP CD4 T cells in RA are unknown, we have found that Egr2 expression in CD4 T cells is induced by TCR stimulation and suppressed by inflammatory cytokines such as IFNγ (Singh et al, 2017), suggesting that the inflammatory condition in patients may repress Egr2 expression in PD-1$^{high}$ MP CD4 T cells which is yet to be investigated.

Egr2/3$^{-/-}$ PD-1$^{high}$ MP CD4 T cells express high levels of chemokine receptors, such as CXCR3, consistent with previous reports for MP T cells (Sallusto et al, 1998). Whether these cells have high propensity for tissue migration and whether this is important for the development of autoimmunity in CD2-Egr2/3$^{-/-}$ mice remains to be investigated.

In addition to the control of inflammatory molecules, molecules such as Myc, P2rx7, and Eomes are regulated by Egr2/3 in PD-1$^{high}$ MP CD4 T cells. These molecules have been found to be important for the homeostatic proliferation of pathogen specific memory T cells (Intlekofer et al, 2005; Bianchi et al, 2006; Borges da Silva et al, 2018). We and others have previously demonstrated that Egr2/3 underpin TCR-mediated proliferation in response to antigen stimulation by promoting expression of regulators of proliferation and enhancing AP-1 signalling (Li et al, 2012; Du et al, 2014; Miao et al, 2017). The similar transcriptional profiles and impairments in proliferation seen in both Egr2/3$^{-/-}$ PD-1$^{high}$ MP CD4 T cells and Egr2/3$^{-/-}$ effector T cells responding to viral infection indicates a general function of Egr2/3 is to support T-cell proliferation.

T-bet is a Th1 regulator and has been found to be highly expressed in PD-1$^{high}$ MP CD4 T cells in the inflamed joints of RA patients (Rao et al, 2017). However, we did not find statistical differences in T-bet expression in PD-1$^{high}$ MP CD4 T cells from peripheral blood between healthy controls and RA patients. This is most likely due to the fact that high levels of T-bet expression were only detected in PD-1$^{high}$ MP CD4 T cells from one third of patients. However, among patients with a high proportion (>10%) of T-bet$^+$ PD-1$^{high}$ MP CD4 T cells, T-bet expression was higher in Egr2$^-$ than

Egr2$^+$ PD-1$^{high}$ MP CD4 T cells. In addition, we previously found that Egr2/3 are suppressors of T-bet function (Singh et al, 2017). Whether suppression of T-bet function in PD-1$^{high}$ MP CD4 T cells is part of the mechanism for Egr2 to control inflammatory autoimmunity in humans is yet to be investigated.

The checkpoint molecule Lag3 is also highly expressed in Egr2$^+$ MP CD4 T cells. Egr2 has been reported to be associated with the function of Lag3$^+$ regulatory T cells (Okamura et al, 2009). However, similar to PD-1, Egr2 is not required for the expression of Lag3. The increased expression of PD-1 and Lag3 in Egr2/3$^{-/-}$ MP CD4 T cells is associated with inflammatory responses indicating that the control mechanisms mediated by Egr2 that regulate inflammatory autoimmune responses differ from those mediated by checkpoint regulators.

Egr2/3 not only control inflammation but also support the homeostatic proliferation of PD-1$^{high}$ MP CD4 T cells and their fitness for adaptive responses to viral infection demonstrating an important function of PD-1$^{high}$ MP CD4 T cells in adaptive immunity. Our findings indicate that disorders of homeostasis of PD-1$^{high}$ MP CD4 T cells can result in both inflammatory autoimmunity and impaired adaptive responses against pathogens. Impaired T-cell receptor–mediated proliferation and hyper-inflammation of MP CD4 T cells have also been found in SLE and RA patients (Cope, 2004; Crispin et al, 2017) further indicating that maintenance of MP T-cell homeostasis is essential for both preventing autoimmunity and supporting adaptive immune responses. The impaired expression of Egr2 in PD-1$^{high}$ MP CD4 T cells from patients with active RA supports the notion that Egr2/3-mediated homeostatic mechanisms play an important part in control of autoimmune responses.

Our findings demonstrate that the Egr2/3-mediated programme is required for the homeostatic fitness of PD-1$^{high}$ MP CD4 cells both to enable their participation in adaptive responses and control auto-immune inflammation, which suggests that modulation of the Egr2/3 programme may provide a new avenue for immune modulation therapy for cancer, chronic infections, and autoimmune diseases.

# Materials and Methods

### Mice

GFP-Egr2 (CD45.2) and CD2-Egr2/3$^{-/-}$ (CD45.2) mice were reported previously (Li et al, 2012; Miao et al, 2017). C57BL/6 (CD45.1) and C57BL/6 (CD45.2) mice were purchased from Charles River and crossed to generate CD45.1/2 mice expressing both allelic variants. All mice analysed were 7–8 wk of age unless otherwise stated. No animal was excluded from the analysis, and the number of mice used was consistent with previous experiments using similar experimental designs. All mice were maintained in the Biological Services Unit, Brunel University, and used according to established institutional guidelines under the authority of a UK Home Office project license.

### Antibodies and flow cytometry

FITC or PE or APC or eFluor450 antibodies to CD4 (clone GK1.5); APC-eFluor780-anti-CD45.1 (clone A20), PEcy7 or APC-anti-IFNγ (clone XMG1.2); PE-antibody to CD3 (clone 145-2C11), APC-anti-CD54 (ICAM-1)

antibody (clone KAT-1), and PerCP-Cy5.5-CXCR (clone CXCR3-173); PE- or PEcy7- or APC-anti-CD25 (clone PC61.5), PE- or PEcy7-anti-CD62L (clone MEL-14), PE- or eFluor450-anti-Ki-67 (clone SolA15), and PE- or APC-antibody to CD45.2 (clone 104); APC or PEcy7-antibodies to CD44 (clone IM7); PE-anti-FOXP3 (clone FJK-16s); APC-anti-T-bet (clone 4B10); and APC or PE-anti-CTLA-4 (clone UC10-4B9) were obtained from eBioscience. PE-anti-mouse CD223 (LAG-3) antibody (clone C9B7W), PE or APC/Cy7-anti-mouse CD279 (PD-1) antibody (clone 29F.1A12), PE-anti-mouse CD5 antibody (clone 53-7.3), BV510-anti-CD44 antibody (clone IM7), PerCP-Cy5.5-anti-CD45.1 antibody (clone A20), APC-anti-CCR5 (clone HM-CCR5) and APC-anti-CD127 (IL-7Rα) (clone A7R34), and Zombie NIR were from BioLegend. APC-labelled MHC/peptide tetramers consisting of H-2 I-A$^b$ MHC molecules bearing OVA$_{329-337}$ or CLIP (control tetramer) were obtained from the National Institutes of Health Tetramer Core facility (Emory University). For staining of human cells, Alexa Fluor 700 anti-CD3 (Cat. no. 317340 clone OKT3), BV510 conjugated anti-CD45RA (Cat. no. 304142 clone HI100), FITC anti-Granzyme B (Cat. no. 515403 clone GB11), BV605 labelled anti-CXCR3 (Cat. no. 353728 clone G025H7), BV711 anti-HLA-DR (Cat. no. 307644 clone L243), and PE-Cy7 anti-PD-1 (Cat. no. 367414 clone NAT105) were purchased from BioLegend, whereas Alexa Fluor 647 anti-T-bet (Cat. no. 561267 clone O4-46), BV421 anti-CD4 (Cat. no. 566392 clone SK3), and BUV395 anti-CD25 (Cat. no. 564034 clone 2A3) were obtained from BD Biosciences. Rabbit anti-Egr2 (Cat. no. ET7108-57 clone JG78-39) was purchased from HuaAn Biotech, whereas PE conjugated F(ab')2-goat anti-rabbit IgG secondary antibody was from eBioscience. Ghost dye 780 was obtained from Tonbo Biosciences. For flow cytometry analysis, single-cell suspensions were analysed on an LSRII, LSRFortessa, or Canto (BD Immunocytometry Systems), and the data were analysed using FlowJo (Tree Star). Cell sorting was performed on a FACSAria sorter with DIVA option (BD Immunocytometry Systems).

## Cell isolation and stimulation

Naïve CD4$^+$ T cells were purified by negative selection using a MACS system (Miltenyi Biotec) or isolated by sorting CD4$^+$CD25$^-$CD44$^{low}$CD62L$^+$ T cells by FACS. MP T cells were isolated by sorting CD4$^+$CD25$^-$CD44$^{high}$CD62L$^-$ cells. GFP-Egr2$^-$ and GFP-Egr2$^+$ MP T cells were isolated by sorting GFP-Egr2$^-$CD4$^+$CD25$^-$CD44$^{high}$CD62L$^-$ and GFP-Egr2$^+$CD4$^+$CD25$^-$CD44$^{high}$CD62L$^-$ cells, respectively. Purified CD4$^+$ T cells were stimulated with plate-bound anti-CD3 at 5 μg/ml (BD Biosciences) and anti-CD28 (2 μg/ml; BD Biosciences) antibodies for 24 h before harvest. MP CD4 T cells were stimulated with 100 ng/ml mouse recombinant IL-12 (BioLegend) for 24 h, or left unstimulated, before analysis of IFNγ-producing cells by intracellular cytokine staining.

For analysis of Egr2, FoxP3, or T-bet expression, the cells were processed using the Foxp3 staining kit (eBioscience). For analysis of cytokine producing cells, the cells were stimulated with 50 ng/ml PMA and 200 ng/ml ionomycin in the presence of Golgistop (BD Biosciences) for 3 h before analysis of cytokine producing cells using the Foxp3 staining kit (eBioscience) and flow cytometry.

## Proliferation

CD44$^{high}$CD4 cells of GFP-Egr2 knock-in (CD45.1) and CD2-Egr2/3$^{-/-}$ (CD45.2) origin isolated from chimeric mice were mixed at a 1:1 ratio and labelled with CellTrace Violet according to the manufacturer's

instructions (Invitrogen). The cells were adoptively transferred to wild-type (CD45.1/2) recipients. Donor cells were analysed by flow cytometry 3 wk after transfer.

## TCRβ sequencing

TCRβ sequencing libraries were generated from three replicate samples of FACS-sorted naïve and MP CD4 T cells from wild-type and Egr2/3$^{-/-}$ mice at 10 wk of age using the SMARTer Mouse TCR a/b Profiling Kit according to the manufacturer's instructions (Clontech). The libraries were sequenced with an Illumina MiSeq platform using a 2 × 300 bp paired-end kit. Base calls, demultiplexing and adapter trimming were performed with Illumina software. Optical duplicates were removed from fastq files using the clumpify function in the BBMAP toolkit (Bushnell, 2018), and sequences were aligned to the IGMT, the international ImMuno-GeneTics information system http://www.imgt.org (founder and director: Marie-Paule Lefranc, Montpellier, France), database of mouse TCRβ genes using the MiXCR algorithm (Bolotin et al, 2015). TCRβ sequences with two or fewer differences with a Phred quality score of 20 or more in all nucleotides were merged into clonotypes using MiXCR and imported into R (R Core Team, 2017) using the tcR package (Nazarov et al, 2015). TCRβ repertoire diversity was visualized using the Shannon entropy normalized to the total number of clonotypes as described (Yohannes et al, 2017). Rank abundance plots were generated using the R package alakazam (Gupta et al, 2015). Briefly, this uses the Chao1 estimator to estimate unseen clonotype numbers with the three replicates for each condition combined. The resulting clonotype estimates are then ranked in the order of clonal size and rank versus clonal size plotted.

## Quantitative real-time PCR

Total RNA was extracted from cells using Trizol (Invitrogen) and reverse transcribed using random primers (Invitrogen). Quantitative real-time PCR was performed on a Rotor-Gene system (Corbett Robotics) using SYBR green PCR master mix (QIAGEN). The primers used are as follows: Myb: sense 5'-CTGAAGATGCTACCTCAGACCC-3' and antisense 5'-TCCCGATTTCTCAGTTGGCG-3'; P2rx7: sense 5'-GACGCTGTGT-CCTGAGTATCC-3' and antisense 5'-GTCATATGGAACACACCTGCC-3'; Tcf7: sense 5'-CCCAGCTTTCTCCACTCTACG-3' and antisense 5'-CTGTG-AACTCCTTGCTTCTGGC-3'; Icam1: sense 5'-GAGCCAATTTCTCATGCC-GC-3' and antisense 5'-AGCTGGAAGATCGAAAGTCCG-3'; and Gapdh: sense 5'-TGCACCACCAACTGCTTAGC-3' and antisense 5'-GGC-ATGGACTGTGGTCATGAG-3'.

The data were analysed using the Rotor-Gene Software. All samples were run in triplicate, and relative mRNA expression levels were obtained by normalizing against the level of Gapdh from the same sample under the same program using: relative expression = 2$^{(CTgapdh - CTtarget)}$.

## RNA-seq analysis

RNA was isolated and purified using TRIzol reagent (Life Technologies). RNA concentration and integrity were assessed using Qubit with an RNA HS reagent kit (Thermo Fisher Scientific) and an Agilent 2100 Bioanalyzer (Agilent Technologies), respectively. Only

RNA samples with RNA integrity values above 7.0 were considered for subsequent analysis. mRNA from T cells from independent biological replicates was processed for directional mRNA-seq library construction using the KAPA mRNA HyperPrep Kit (Roche Sequencing Solutions) according to the manufacturer's protocol. We performed 43-nt paired-end sequencing using an Illumina NextSeq 500 platform. Base calls, demultiplexing and adapter trimming were performed with Illumina software. The short sequenced reads were mapped to the mm10 build of the mouse reference genome using the spliced aligner Hisat2 (Kim et al, 2015). Intermediate processing steps to remove secondary alignments and pairs where only one read was mapped were performed using SAMtools (Li et al, 2009), whereas optical duplicates were removed with Picard (Broad Institute, 2016). We used several R/Bioconductor (R Core Team, 2017) packages to identify genes differentially expressed between GFP-Egr2$^+$ and GFP-Egr2$^-$ or Egr2/3$^{-/-}$ T cells. Briefly, the number of reads mapped to each gene on the basis of the UCSC refGene database (available from https://genome.ucsc.edu/) were counted, reported, and annotated using the BiocParallel, Rsamtools, GenomicAlignments, GenomicFeatures, and org.Mm.eg.db packages (Lawrence et al, 2013; Carlson, 2017; Morgan et al, 2017a, 2017b). To identify genes differentially expressed between groups, we used the R/Bioconductor package edgeR (Robinson et al, 2010). Briefly, count data were first normalized and dispersion estimated before a negative binomial model was fitted with significance assessed by a quasi-likelihood F-test (Lun et al, 2016). Resulting *P*-values were adjusted for multiple testing using the Benjamini–Hochberg procedure. Genes with an adjusted *P*-value less than or equal to 0.05 and an absolute fold change greater than or equal to 1.5 were considered differentially expressed.

For the heat map, a variance stabilizing transformation from the DESeq2 and vsn packages (Huber et al, 2002; Love et al, 2014) was applied to the dataset and selected genes were "row-centred" by subtraction of the mean expression level for each gene before hierarchical clustering and visualization with the ComplexHeatmap package (Gu et al, 2016).

For functional annotation, the msigdbr package (Dolgalev, 2018) was used to obtain Mouse Entrez Gene IDs corresponding to the Broad Institute Hallmark gene sets (Liberzon et al, 2015). For Gene Set Enrichment-type analysis, data were processed using the voom with quality weights methodology in the limma package (Liu et al, 2015; Ritchie et al, 2015) to generate normally distributed data and then mean ±95% confidence intervals and enrichment *P*-values for each gene set were calculated using the qusage package (Yaari et al, 2013). For the volcano plots the Benjamini–Hochberg corrected *P*-values and log$_2$ fold changes, calculated from the edgeR data, the total dataset was plot using the ggplot2 package (Wickham, 2016) and then selected genes were highlighted.

## ChIP and ChIP-seq assays

ChIP-seq assays were performed according to published methods (Kidder & Zhao, 2014). Briefly, 5 × 10$^7$ CD4 cells from GFP-Egr2 mice were stimulated with anti-CD3 and anti-CD28 for 24 h. The cells were then cross-linked with 1% formaldehyde for 10 min at room temperature. After quenching of formaldehyde with 125 mM glycine, chromatin was sheared by sonication with a Bioruptor Pico sonication system

(Diagenode). The fragmented chromatin was around 200–500 bp as analysed on agarose gels. After preclearing, chromatin (500 μg) was subjected to immunoprecipitation with GFP-Trap MA (Chromotek), or anti-Egr2 polyclonal antibody (Covance), or Ig as negative control, bound to blocked protein G beads at 4°C overnight. DNA was purified by phenol chloroform extraction and concentration was measured by Qubit with a dsDNA HS assay kit (Thermo Fisher Scientific).

For validation of a successful IP, ChIP DNA was used as template for PCR amplification in triplicate with specific primers flanking the Egr2 binding sites (Miao et al, 2017). The primers used are as follows: *Nab2* sense 5′-GAGAGGCTGCTGTGGAGACT-3′ and antisense 5′-GTACGTGGGCGCAGAGAG-3′; *Tcf7* sense 5′-CAACGCATGTGATCACC-CACC-3′ and antisense 5′-TCCTGAAAGAAGAGGCGTCCG-3′. Data are expressed as the percentage of input DNA recovered.

For ChIP-seq, libraries from three independent IPs were generated using the NEBNext Ultra II DNA Library Prep kit according to the manufacturer's instructions. We performed 75 bp single-end sequencing using an Illumina NextSeq 500 platform. Base calls, demultiplexing, and adapter trimming were performed with Illumina software. The short sequenced reads were mapped to the mm10 build of the mouse reference genome using Bowtie2 (Langmead & Salzberg, 2012). Intermediate processing steps to remove secondary alignments and alignments with a MAPQ < 30 were performed using SAMtools (Li et al, 2009), whereas duplicates were removed with Picard (Broad Institute, 2016). To generate high confidence peaks, the IDR methodology (Li et al, 2011; Landt et al, 2012) was used using spp (Kharchenko et al, 2008) for cross-correlation analysis and peak calling and IDR version 2 for subsequent analysis. Peaks were annotated using the ChIPpeakAnno and ChIPseeker packages (Zhu et al, 2010; Yu et al, 2015), whereas functional enrichment was performed using a hypergeometric test, as implemented in the clusterProfiler package (Yu et al, 2012), with Broad Institute Hallmark gene sets (Liberzon et al, 2015). Motif analysis was performed using homer (Heinz et al, 2010). ChIP-seq and RNA-seq tracks were generated using deeptools (Ramírez et al, 2016) and visualized using IGV (Thorvaldsdóttir et al, 2013).

## Bone marrow chimeras and OT-II retrogenic mice

Bone marrow was collected from CD2-Egr2/3$^{-/-}$ (CD45.2+) or GFP-Egr2 (CD45.1+) mice. For each chimera, 10 × 10$^6$ cells of a 1:1 mixture of CD2-Egr2/3$^{-/-}$ and GFP-Egr2 bone marrow cells were transferred intravenously into lethally irradiated (two doses of 550 rad) wild-type C57BL/6 (CD45.1/2) recipients. For OT-II retrogenic mice, the OT-II-2A.pMIG II construct, a kind gift from Dario Vignali (plasmid #52112; Addgene; http://n2t.net/addgene:52112; RRID:Addgene_52112), (Holst et al, 2006a, 2006b), was transfected into Phoenix cells (Clontech) as described (Zhu et al, 2008). Bone marrow cells isolated from CD2-Egr2/3$^{-/-}$ and wild-type C57BL/6 mice were cultured with IL-3, IL-6, and SCF (BioLegend) and transduced with retroviral supernatant from transfected Phoenix cells by spin transduction as described (Holst et al, 2006a; Bettini et al, 2013). The transduced cells were analysed for expression of GFP by flow cytometry. If more than 5% of cells were GFP$^+$, the cells were transferred into lethally irradiated (two doses of 550 rad) wild-type C57BL/6 recipients as described (Holst et al, 2006a; Bettini et al, 2013). Recipient mice were allowed 8–12 wk for reconstitution.

## Adoptive transfer

Wild-type OT-II (CD45.1$^+$GFP$^+$CD4$^+$CD44$^{high}$) and Egr2/3$^{-/-}$ OT-II (CD45.2$^+$GFP$^+$CD4$^+$CD44$^{high}$) MP cells were isolated from OT-II retrogenic mice by FACS, and expression of OT-II TCR was confirmed by staining with APC-labelled I-A$^b$-OVA$_{329-337}$ tetramer. 3 × 10$^5$ wild-type and 3 × 10$^5$ Egr2/3$^{-/-}$ retrogenic OT-II MP cells were mixed and adoptively transferred to C57BL/6 mice (CD45.1/2). In half of the recipient mice, the donor cells were quantified and phenotypically analysed 24 h or 3 wk after transfer. For the other half, the recipient mice were infected i.p. with 2 × 10$^5$ PFU of vaccinia virus (OVA-VV$_{WR}$) as described in our report (Miao et al, 2017). 7 d after infection, donor cells were quantified and phenotypically analysed.

### Human study

Research involving human subjects was performed according to the guidelines from the Local Ethical Review Committee, Dong Fang hospital, Beijing Chinese Medicine University through approved protocols with appropriate informed consent obtained. Patients with RA fulfilled the ACR 2010 RA classification criteria. C-reactive protein level and medication usage were obtained by review of digital medical records (Table S1). Biological therapy was defined as the use of anti-TNF, abatacept, rituximab, tocilizumab, or tofacitinib. All blood samples were obtained from RA patients seen at the Dong Fang hospital Arthritis Center, Dong Fang hospital, Beijing Chinese Medicine University. Blood samples were acquired before initiation of a new biological therapy or within 1 wk of starting methotrexate. Peripheral blood mononuclear cells (PBMCs) were isolated from blood using Ficoll-Paque (Sigma-Aldrich) according to the manufacturer's protocol. All blood CD4$^+$ T cell analyses focussed on CD45A$^-$ memory (MP) CD4 T cells which includes both resting and activated MP cells. The non-inflammatory healthy controls were staff at Dong Fang hospital, Beijing Chinese Medicine University.

### Statistics

To analyse the statistical significance of differences between groups, two-tailed Mann–Whitney tests using the R package coin (Hothorn et al, 2008) or Kruskal–Wallis tests followed by pairwise comparisons using Conover tests, as implemented in the R package PMCMRplus (Pohlert, 2018), with Benjamini–Hochberg correction for multiple comparisons were used as indicated. Student's unpaired two-tailed $t$ tests were used for in vitro experiments. Differences with a $P$-value < 0.05 were considered significant.

## Data Availability

RNA-seq and ChIP-seq data are available from ArrayExpress under accession numbers E-MTAB-7795 and E-MTAB-7797, respectively, whereas TCR-seq data are available from the European Nucleotide Archive under study number PRJEB33211.

## Supplementary Information

## Acknowledgements

We thank Dr Miguel Branco, Dr Ozgen Deniz, and Dr Ben Kidder for advice on ChIP-seq and the National Institutes of Health Tetramer Core facility (Emory University, Atlanta, GA) for providing APC-labelled I-A$^b$-OVA$_{329-337}$ and I-A$^b$-CLIP tetramers. We thank UCL Genomics, University College London, UK, for RNA-seq library preparation and sequencing and the Barts and the London Genome Centre, Queen Mary University of London, for sequencing of ChIP-seq and TCR-seq libraries. This work was supported by the Medical Research Council, UK (MR/N00096X/1), Barts Charity (MGU0463), and National Science Foundation of China (81774275).

### Author Contributions

ALJ Symonds: conceptualization, formal analysis, investigation, methodology, and writing—review and editing.
W Zheng: investigation.
T Miao: conceptualization, resources, formal analysis, investigation, methodology, and writing—review and editing.
H Wang: investigation.
T Wang: investigation.
R Kiome: investigation.
X Hou: conceptualization, resources, funding acquisition, investigation, and writing—review and editing.
S Li: conceptualization, resources, formal analysis, funding acquisition, investigation, and writing—review and editing.
P Wang: conceptualization, funding acquisition, investigation, and writing—original draft, review, and editing.

### Conflict of Interest Statement

The authors declare that they have no conflict of interest.

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
