## [Reviewer comments · Life Science Alliance]

Life Science Alliance

Egr2 and 3 control inflammation, but maintain homeostasis, of PD-1^{high} memory phenotype CD4 T cells

Alistair Symonds, Wei Zheng, Tizong Miao, Haiyu Wang, TieShang Wang, Ruth Kiome, Xiujuan Hou, Suling Li, and Ping Wang

DOI: <https://doi.org/10.26508/lsa.202000766>

Corresponding author(s): Ping Wang, Barts and The London School of Medicine and Dentistry, Queen Mary University of London; Suling Li, Brunel University London; and Xiujuan Hou, Dongfang Hospital, Beijing University of Chinese Medicine

Review Timeline:

Submission Date:	2020-05-05
Editorial Decision:	2020-05-25
Revision Received:	2020-07-01
Editorial Decision:	2020-07-03
Revision Received:	2020-07-09
Accepted:	2020-07-10

Transaction Report:

May 25, 2020

Re: Life Science Alliance manuscript #LSA-2020-00766-T

Prof. Ping Wang
Barts and The London School of Medicine and Dentistry, Queen Mary University of London
The Blizard Institute
4 Newark Street
London E1 2AT
United Kingdom

Dear Dr. Wang,

Thank you for submitting your manuscript entitled "Egr2 and 3 control inflammation, but maintain homeostasis, of PD-1high memory phenotype CD4 T cells" to Life Science Alliance. The manuscript was assessed by expert reviewers, whose comments are appended to this letter.

As you will see, the reviewers appreciate your analysis and provide constructive input on how to further strengthen your manuscript. We would thus like to invite you to submit a revised version of your manuscript to us, addressing the individual points raised.

Thank you for this interesting contribution to Life Science Alliance. We are looking forward to receiving your revised manuscript.

Sincerely,

Andrea Leibfried, PhD

Executive Editor
Life Science Alliance
Meyerhofstr. 1
69117 Heidelberg, Germany
t +49 6221 8891 414
e contact@life-science-alliance.org
www.life-science-alliance.org

B. MANUSCRIPT ORGANIZATION AND FORMATTING:

Reviewer #1 (Comments to the Authors (Required)):

Symonds et al detail the effects of Egr2/3 expression on homeostasis, gene expression, and cytokine production by memory-phenotype (MP) PD1+CD4+ T cells. Their work nicely shows that the expression of Egr2/3, while not essential for generation of the PD1+ (and Lag3+) phenotype, does have an important role in both promoting survival/proliferation and suppressing cytokine secretion by these MP T cells. The latter effects may in part be due to effects of Egr2/3 on

regulation of Tbet expression. Overall, the work is convincing and seems to be well controlled and presented. However, a few clarifications/additions would improve accessibility or impact of the work.

Specific comments:

Fig1E is cited in 1st section of results but there is not panel E in fig 1. Also, it seems unsurprising that only mice with the Egr2-gfp cells stain positive for Egr2-gfp (Fig. 1D). Unclear what the point of this panel is.

Fig 2 states that Ki67 staining was evaluated in mice that were 7 weeks old. How long was this after reconstitution? Also, the transfers were into wt mice. What is driving the homeostatic proliferation here?

Fig 3B - the color scale for P values here is not legible. Suggest labeling rows individually. Is there differential expression of IL2/15Rb? Perhaps this could explain the effects on proliferation/homeostasis.

Fig 5 J/K indicate increased IFN γ production by EGR2/3^{-/-} OTII cells after infection and data in Fig 7 further support a suppressive effect of Egr2/3 expression of IFN γ in response to IL-12. Presumably the effects of Egr 2/3 on IFN γ are secondary to effects on tbet (Tbx21). However, only a subset of the Egr2/3⁻ MP T cells are producing IFN γ . Are the remaining T cells not responding to IL-12 (IL12R or STAT4 neg)? Do they instead produce more IL-10 (expected from RNA) or other effector cytokines (IL-17, IL-4, etc) during infection, in response to IL-12, or when stimulated in other contexts?

It seems important to also comment on whether the proportion of FoxP3⁺ CD4 T cells reduced in the Egr2/3⁻ mice, which might also promote an inflammatory response.

Reviewer #2 (Comments to the Authors (Required)):

In the manuscript "Egr2 and 3 control inflammation, but maintain homeostasis, of PD-1^{high} memory phenotype CD4 T cells", Symonds and Zheng et al investigate the mechanism by which Egr2 and 3 control autoimmune activity by memory phenotype (MP) cells. They demonstrate, using GFP-Egr2 knockin and Egr2/3 conditional knockout mice and an OT-II retrogenic model, that while Egr2 and Egr3 are not required for formation of PD1^{high} MP cells, they are required for proliferation and homeostasis of these cells, promote a distinct gene expression pattern, and are required to control inflammatory responses by these cells. They also analyzed these cells in rheumatoid arthritis patients and found reduced Egr2 associated with inflammatory responses in RA.

The manuscript is clearly written. A major strength of this study is use of several genetically modified mouse lines and mixed bone marrow chimeras to ensure study of cell-intrinsic effects of EGR2 on PD1^{hi} MP cells. Another strength is correlation of these results to data from human patients. These findings take important steps towards elucidating the mechanism by which MP cells contribute to autoimmunity. However, there is some concern regarding the use of Egr2/3 double conditional knockout as a control for the gene expression analysis. Specifically, Egr2 and Egr3 have distinct roles in T cells as evidenced in part by a previous publication in which more severe autoimmunity was found in the absence of both (Morita et al 2016). As such, it is not clear why the authors chose to determine EGR2 regulated genes using the double knockout for the

RNAseq portion of the analysis rather than an EGR2 knockout and how to interpret these genes as being directly regulated specifically by EGR2.

Additional minor issues:

1. The conclusion that Egr2/3 support self-renewal of PD-1^{high} MP cells appears to be based primarily on the fact that Egr2 levels remain constant with cell divisions. It would be more appropriate to either modify the conclusion to focus on homeostatic expansion OR include further evidence that the population after cell divisions is the same cell population. Specifically, data demonstrating the same phenotype across multiple markers for the undivided and most divided peaks would provide further evidence towards this point.
2. For differential gene expression in Fig. 3A, the list of genes listed in the text description includes genes not listed on the figure. It would help the clarity for the reader if the list of genes in the text and in the figure were the same.
3. The discussion of Egr2/3 targets and genes associated with proliferation defects in results for Fig. 4E,F does not specify which genes are upregulated or downregulated with proliferation defects. Detail on upregulation vs downregulation would add clarity for the reader.
4. It is not entirely clear what each of the top three rows for each gene in Figure 4F represent - they are clearly chromatin peaks, but it's unclear what the comparison is between the three. And how are RNA-seq reads incorporated in this figure?
5. When Figure S2 is referenced, clarity would be enhanced by citing the subsections for each point of discussion.
6. Figure 6 depicts analysis of CD127 through representative data. Providing the data across all samples would provide statistical justification for your conclusion.
7. The analysis in Figure 6 and accompanying text for CD127 levels suggest that the cells respond to IL7 based on receptor expression, but this is not proven. It would improve this section to either temper the conclusion so that response to IL-7 is not the focus, or to complete an experiment stimulating the cells in vitro through CD127 and measuring activation of downstream signaling.
8. Did the healthy controls in Fig. 8C,D also have >10% T-bet⁺? If not, how were they selected? More broadly, to the reader Fig. 8A, B are very strong points that are well suited to a strong ending. C and D may dilute the point, and therefore might be more appropriate as supplemental data.

Reviewer #3 (Comments to the Authors (Required)):

Symonds et al provide a manuscript describing the roles of EGR2/3 on memory T cell function with a focus on PD-1^{hi} CD4⁺ T cells. They demonstrate Egr2/3 expression in a Cd4⁺ memory T cell population enriched for memory and Th1 features. These cells appear proliferative with a distinctive transcriptomic signature. In parallel, Egr2/3^{-/-} cells are found to be abnormal with increased PD1 expression, decreased proliferative capacity, reduced TCR repertoire, and increased Th1 skewing. In RA patients, the expanded PD-1^{hi} T cell population shows reduced Egr2/3 expression, associated with increased Tbet and GZMB expression. The work highlights Egr2/3 as an interesting regulator of CD4 T cell proliferation and function, now placed in the context of pathologically expanded PD1 high T cells in autoimmunity. The work in multiple models, using complementary approaches, with in vivo experiments, transcriptomics, and ChIP analyses, is broad and interesting. It would be valuable for the authors to address these points, in particular with focus on clarifying 1) the relationship between 'PD1^{hi}' cells studied here and the PD1^{hi} cells described in autoimmune patients, and 2) the extent to which PD1⁺ cells in Egr2/3^{-/-} mice can be compared to the Egr2/3⁺ PD1⁺ subset in WT or GFP⁺ mice.

MAJOR COMMENTS:

1) PD1 expression:

How do the authors distinguish 'PD-1hi' versus PD1+ or PD1intermediate? Figure 1 only indicates a PD1 +/- cutoff. Is this the same gate used to analyze cells in Figure? Does Hi = positive?

2) Experimental numbers: Figure 1,2,

The total numbers of mice should be indicated more clearly.

For example, Figure 1C/D. How many mice are analyzed here? The legend indicates 3 experiments, but there are 4 data points. Are there multiple mice per group per experiment?

3) Egr2/3-/- cells

It is not clear how to think about the T cells in the Egr2/3-/- mice. They all express PD1, yet they look like functionally and transcriptomically seem to resemble PD-1- EGR2/3- cells in WT mice. Do the authors have evidence to suggest the PD1+ cells in Egr2/3-/- are phenotypically like PD-1hi cells described in disease (e.g. the Tfh-like cells in Rao/Zhang/Bocharnikov)? The authors could discuss further what they think of this T cell phenotype in the absence of Egr2/3. They seem unlikely to be the same T cell population as the PD1hi subset in WT mice.

4) The authors have not clearly demonstrated that PD1hi cells they analyze are functionally similar to PD1hi Tfh-like cells in RA patients (studies they reference). It still seems possible that the PD1hi cells studied in this report are Tregs or other activated cells unrelated to Tfh cells. Can the authors demonstrate a Tfh-like function, or specific enrichment in Tfh-like features, in the 'PD1hi' cells they study?

5) Egr+ cells

The transcriptomics suggest high expression of FoxP3 and CD25 in the Egr2/3+ cells. Are they enriched Tregs? If Egr is induced by TCR activation, then are the Egr+ cells in the GFP mice largely an activated T cell population?

6) In the adoptive transfer experiments with OTII mice, did the authors evaluate if Egr2/3 loss changes the migration of the cells, such that they migrate to different tissues? differential expression of CCR9, ICAM, etc could reflect changes in migratory capacity.

MINOR:

1) the "MP" abbreviation is non-standard and not that helpful. The authors could describe the cells as a memory T cell subset and then omit the MP label throughout the manuscript to improve readability, or simply use the term 'memory'.

2) It appears that all CD44hi cells are CCR5+ and the majority are CXCR3+. This seems unexpected; how do the authors explain this staining pattern?

3) It is not clear what the authors mean with the term 'virtual' MP T cells or 'virtual' PD-1hi cells

Editor

Life Science Alliance

Dear Editor

We would like to thank the reviewers for their valuable comments and suggestions regarding our manuscript (LSA-2020-00766-T). We have addressed these points and added additional data into the revised manuscript. Our responses to the comments are as follows:

Reviewer #1 (Comments to the Authors (Required)):

Symonds et al detail the effects of Egr2/3 expression on homeostasis, gene expression, and cytokine production by memory-phenotype (MP) PD1+CD4+ T cells. Their work nicely shows that the expression of Egr2/3, while not essential for generation of the PD1+ (and Lag3+) phenotype, does have an important role in both promoting survival/proliferation and suppressing cytokine secretion by these MP T cells. The latter effects may in part be due to effects of Egr2/3 on regulation of Tbet expression. Overall, the work is convincing and seems to be well controlled and presented. However, a few clarifications/additions would improve accessibility or impact of the work.

Specific comments:

Fig1E is cited in 1st section of results but there is not panel E in fig 1. Also, it seems unsurprising that only mice with the Egr2-gfp cells stain positive for Egr2-gfp (Fig. 1D). Unclear what the point of this panel is.

The erroneous reference to Fig 1E has now been removed. We previously showed that Egr2 expression in effector T cells is transient and abolished upon transfer to uninfected mice (Miao et al., 2017). In contrast, Fig 1D shows that Egr2 expression in MP T cells is maintained upon adoptive transfer. We have now expanded upon this in the text to emphasize this point (page 5).

Fig 2 states that Ki67 staining was evaluated in mice that were 7 weeks old. How long was this after reconstitution? Also, the transfers were into wt mice. What is driving the homeostatic proliferation here?

Ki67 was analysed between 8 to 12 weeks after reconstitution. This information has been added to the figure legend. In wild type mice, homeostatic proliferation is largely driven by homeostatic cytokines such as IL-7 (Boyman et al., 2007; Raeber et al., 2018).

Fig 3B - the color scale for P values here is not legible. Suggest labeling rows individually. Is there differential expression of IL2/15Rb? Perhaps this could explain the effects on proliferation/homeostasis.

Fig 3B has been improved to make it clearer. Il2rb expression is not altered in K23 or GFPlo cells compared to GFPHi in the RNAseq.

Fig 5 J/K indicate increased IFNg production by EGR2/3-/- OTII cells after infection and data in Fig 7 further support a suppressive effect of Egr2/3 expression of IFNg in response to IL-12. Presumably the effects of Egr 2/3 on IFNg are secondary to effects on tbet (Tbx21). However, only a subset of the Egr2/3- MP T cells are producing IFNg. Are the remaining T cells not responding to IL-12 (IL12R or

STAT4 neg)? Do they instead produce more IL-10 (expected from RNA) or other effector cytokines (IL-17, IL-4, etc) during infection, in response to IL-12, or when stimulated in other contexts?

We and others previously found that only a subset of T cells produce IFN γ during infection (e.g. Miao et al 2017, Roman et al 2002, Foulds and Shen 2006). The reasons for this are unclear but may reflect different microenvironments. As the reviewer points out, the effects on Ifn γ are likely secondary to effects on T-bet as we published previously (Singh et al., 2017). As we have shown in Fig. 7C and D, around 40-45% of Egr2/3^{-/-} MP cells express T-bet while the expression of Ifn γ by Egr2/3^{-/-} MP during infection is only slightly lower (30-40%). We did not find differential expression of Th17 and Th2 cytokines in the RNAseq. Although it will be interesting to investigate the differences between Egr2^{-/-} and Egr2⁺ MP in cytokine production and other functions under different conditions, in this study we are focusing on homeostatic conditions.

It seems important to also comment on whether the proportion of FoxP3⁺ CD4 T cells reduced in the Egr2/3⁻ mice, which might also promote an inflammatory response.

We previously showed that Treg numbers and function were unchanged in Egr2/3^{-/-} mice (Li et al 2012). In addition, we have now added data in to the revised manuscript showing that the proportions of Tregs amongst Egr2/3^{-/-} and Egr2⁺ CD44^{high} cells are similar (Fig 1B)

Reviewer #2 (Comments to the Authors (Required)):

In the manuscript "Egr2 and 3 control inflammation, but maintain homeostasis, of PD-1^{high} memory phenotype CD4 T cells", Symonds and Zheng et al investigate the mechanism by which Egr2 and 3 control autoimmune activity by memory phenotype (MP) cells. They demonstrate, using GFP-Egr2 knockin and Egr2/3 conditional knockout mice and an OT-II retrogenic model, that while Egr2 and Egr3 are not required for formation of PD1^{high} MP cells, they are required for proliferation and homeostasis of these cells, promote a distinct gene expression pattern, and are required to control inflammatory responses by these cells. They also analyzed these cells in rheumatoid arthritis patients and found reduced Egr2 associated with inflammatory responses in RA.

The manuscript is clearly written. A major strength of this study is use of several genetically modified mouse lines and mixed bone marrow chimeras to ensure study of cell-intrinsic effects of EGR2 on PD1^{hi} MP cells. Another strength is correlation of these results to data from human patients. These findings take important steps towards elucidating the mechanism by which MP cells contribute to autoimmunity. However, there is some concern regarding the use of Egr2/3 double conditional knockout as a control for the gene expression analysis. Specifically, Egr2 and Egr3 have distinct roles in T cells as evidenced in part by a previous publication in which more severe autoimmunity was found in the absence of both (Morita et al 2016). As such, it is not clear why the authors chose to determine EGR2 regulated genes using the double knockout for the RNAseq portion of the analysis rather than an EGR2 knockout and how to interpret these genes as being directly regulated specifically by EGR2.

We and others have previously demonstrated that Egr2 and 3 have overlapping functions in T cells, but Egr2 is dominant in all the functions analysed (Li et al 2012, Morita et al 2016, Miao et al 2017). Autoimmunity is not observed in Egr3 knockout while CD2-Egr2^{-/-} only develops autoimmunity in later life (Tourtellotte et al 1998, Zhu et al 2008, Li et al 2012). However, the severe autoimmunity and loss of homeostasis of T cells is only seen in Egr2/3^{-/-} mice (Zhu et al, 2008; Li et al, 2012; Morita et al 2016) as the reviewer points out. Therefore, we utilised Egr2/3^{-/-} cells for RNAseq to avoid compensation by Egr3.

Additional minor issues:

1. The conclusion that Egr2/3 support self-renewal of PD-1^{high} MP cells appears to be based primarily on the fact that Egr2 levels remain constant with cell divisions. It would be more appropriate to either modify the conclusion to focus on homeostatic expansion OR include further evidence that the population after cell divisions is the same cell population. Specifically, data demonstrating the same phenotype across multiple markers for the undivided and most divided peaks would provide further evidence towards this point.

We have now changed the text to refer to homeostatic proliferation rather than self-renewal (pages 5, 6, 15, 17).

2. For differential gene expression in Fig. 3A, the list of genes listed in the text description includes genes not listed on the figure. It would help the clarity for the reader if the list of genes in the text and in the figure were the same.

We have now improved Fig 3A to better reflect the text as the reviewer suggested.

3. The discussion of Egr2/3 targets and genes associated with proliferation defects in results for Fig. 4E,F does not specify which genes are upregulated or downregulated with proliferation defects. Detail on upregulation vs downregulation would add clarity for the reader.

We have now added this to the text to make it clearer (page 8).

4. It is not entirely clear what each of the top three rows for each gene in Figure 4F represent - they are clearly chromatin peaks, but it's unclear what the comparison is between the three. And how are RNA-seq reads incorporated in this figure?

The first two rows are the RNAseq reads for that gene for Egr2⁺ and Egr2/3^{-/-} MP cells, while the third is the peaks from GFP-Egr2 ChIPseq. We have now improved the figure legend to make this clearer.

5. When Figure S2 is referenced, clarity would be enhanced by citing the subsections for each point of discussion.

Citation of subsections has now been added to the text (page 9).

6. Figure 6 depicts analysis of CD127 through representative data. Providing the data across all samples would provide statistical justification for your conclusion.

We have now performed a statistical analysis of CD127 which shows that the percentage of CD127^{high} cells is similar between Egr2⁺ MP and Egr2/3^{-/-} MP. These data have now been included in revised Fig 6B.

7. The analysis in Figure 6 and accompanying text for CD127 levels suggest that the cells respond to IL7 based on receptor expression, but this is not proven. It would improve this section to either temper the conclusion so that response to IL-7 is not the focus, or to complete an experiment stimulating the cells in vitro through CD127 and measuring activation of downstream signaling.

We have now reworded this section accordingly (page 10).

8. Did the healthy controls in Fig. 8C,D also have >10% T-bet+? If not, how were they selected? More broadly, to the reader Fig. 8A, B are very strong points that are well suited to a strong ending. C and D may dilute the point, and therefore might be more appropriate as supplemental data.

The healthy controls in Fig 8C had percentages of T-bet+ cells close to the average for the healthy controls (~5%). The statistical data in Fig 8D are comparing T-bet expression in Egr2+ and Egr2- cells from patients only. We have now improved the figure legend to make this clearer. We think antagonising T-bet function is one of the major roles of Egr2. Therefore, we will present these in our main text.

Reviewer #3 (Comments to the Authors (Required)):

Symonds et al provide a manuscript describing the roles of EGR2/3 on memory T cell function with a focus on PD-1hi CD4+ T cells. They demonstrate Egr2/3 expression in a Cd4+ memory T cell population enriched for memory and Th1 features. These cells appear proliferative with a distinctive transcriptomic signature. In parallel, Egr2/3-/- cells are found to be abnormal with increased PD1 expression, decreased proliferative capacity, reduced TCR repertoire, and increased Th1 skewing. In RA patients, the expanded PD-1hi T cell population shows reduced Egr2/3 expression, associated with increased Tbet and GZMB expression. The work highlights Egr2/3 as an interesting regulator of CD4 T cell proliferation and function, now placed in the context of pathologically expanded PD1 high T cells in autoimmunity. The work in multiple models, using complementary approaches, with in vivo experiments, transcriptomics, and ChIP analyses, is broad and interesting. It would be valuable for the authors to address these points, in particular with focus on clarifying 1) the relationship between 'PD1hi' cells studied here and the PD1hi cells described in autoimmune patients, and 2) the extent to which PD1+ cells in Egr2/3-/- mice can be compared to the Egr2/3+ PD1+ subset in WT or GFP+ mice.

MAJOR COMMENTS:

1) PD1 expression:

How do the authors distinguish 'PD-1hi' versus PD1+ or PD1intermediate? Figure 1 only indicates a PD1 +/- cutoff. Is this the same gate used to analyze cells in Figure? Does Hi = positive?

Yes, PD-1high is the same as PD-1+ and the same gate was used as in this figure.

2) Experimental numbers: Figure 1,2,

The total numbers of mice should be indicated more clearly.

For example, Figure 1C/D. How many mice are analyzed here? The legend indicates 3 experiments, but there are 4 data points. Are the multiple mice per group per experiment?

Three experiments with four to five mice in each group in each experiment. The presented data are from four mice from one experiment.

3) Egr2/3-/- cells

It is not clear how to think about the T cells in the Egr2/3-/- mice. They all express PD1, yet they look like functionally and transcriptomically seem to resemble PD-1- EGR2/3- cells in WT mice. Do you the authors have evidence to suggest the the PD1+ cells in Egr2/3-/- are phenotypically like PD-1hi cells described in disease (e.g. the Tfh-like cells in Rao/Zhang/Bocharnikov)? The authors could discuss further what they think of this T cell phenotype in the absence of Egr2/3. They seem unlikely to be the same T cell population as the PD1hi subset in WT mice.

The papers showing enrichment of pathogenic PD-1hi T cells in RA and SLE have found that these cells do not express the Tfh marker CXCR5 but instead express CXCR3 and CCR5 (Rao et al, 2017; Zhang et al, 2019; Bocharnikov et al, 2019; Caielli et al, 2019). These markers are highly expressed by Egr2+ MP cells and also by Egr2/3-/- MP cells (Fig. 1B). In terms of these cell surface markers, Egr2/3-/- MP cells more closely resemble Egr2+ MP cells than Egr2- MP cells (Fig. 1B). We hypothesise that this population is normally controlled by Egr2 and the absence of Egr2 and 3 in the Egr2/3-/- cells leads to them acquiring some functional and transcriptomic features of the Egr2- MP population. This hypothesis will be investigated further in future studies.

4) They authors have not clearly demonstrated the that PD1hi cells they analyze are functionally similar to PD1hi Tfh-like cells in RA patients (studies they reference). It still seems possible that the PD1hi cells studied in this report are Tregs or other activated cells unrelated to Tfh cells. Can they authors demonstrate a Tfh-like function, or specific enrichment in Tfh-like features, in the 'PD1hi' cells they study?

The pathogenic PD-1hi T cells which have been discovered in RA and SLE are distinct from Tfh cells and do not express the Tfh marker CXCR5; instead they express CXCR3 and CCR5 along with high levels of Il21 and Il10 (Rao et al, 2017; Zhang et al, 2019; Bocharnikov et al, 2019; Caielli et al, 2019). These markers and cytokines are highly expressed by Egr2/3-/- MP cells (Fig. 1B, Fig 3A). This point is highlighted in the discussion (page 15).

5) Egr+ cells

The transcriptomics suggest high expression of FoxP3 and CD25 in the Egr2/3+ cells. Are they enriched Tregs? If Egr is induced by TCR activation, then are the Egr+ cells in the GFP mice largely an activated T cell population?

We have now added data showing that although some CD44high Egr2+ cells are indeed Tregs, the majority are not and, importantly, the Egr2/3-/- CD44high population had a similar proportion of Tregs (Fig 1B).

Egr2 is indeed induced by TCR activation in vitro and during viral infection but the Egr2+ MP cells lack expression of activation markers such as CD25.

6) In the adoptive transfer experiments with OTII mice, did the authors evaluate if Egr2/3 loss changes the migration of the cells, such that they migrate to different tissues? differential expression of CCR9, ICAM, etc could reflect changes in migratory capacity.

We did not analyse the migration of OT-II cells. However, in our previous experiments with OT-I effector T cells we found that the numbers of cells in infected tissues were also reduced indicating that the observed impairments in expansion are not due to increased migration (Miao et al 2017).

MINOR:

1) the "MP" abbreviation is non-standard and not that helpful. The authors could describe the cells as a memory T cell subset and then omit the MP label throughout the manuscript to improve readability, or simply use the term 'memory'.

The term MP has been used by several groups (e.g. Younes et al., 2011 PLoS Biol 2011 Oct;9(10):e1001171; Su et al., 2013, Immunity. 2013 Feb 21;38(2):373-83; Kawabe et al., 2017 Sci Immunol. 2017 Jun 16;2(12)).

2) It appears that all CD44hi cells are CCR5+ and the majority are CXCR3+. This seems unexpected; how do the authors explain this staining pattern?

CCR5 and CXCR3 are known to be preferentially expressed by MP cells (Bleul et al 1997, Oberbarnscheidt et al 2011, Li et al 2016) and it has been previously reported that the majority of MP cells are CXCR3+ (Sallusto et al 1998). We do not know why the levels of these markers in our mice are higher than some reports.

3) It is not clear what the authors mean with the term 'virtual' MP T cells or 'virtual" PD-1hi cells

This term was used for the OT-II cells since these cells have been generated in an antigen independent fashion.

July 3, 2020

RE: Life Science Alliance Manuscript #LSA-2020-00766-TR

Prof. Ping Wang
Barts and The London School of Medicine and Dentistry, Queen Mary University of London
The Blizard Institute
4 Newark Street
London E1 2AT
United Kingdom

Dear Dr. Wang,

Thank you for submitting your revised manuscript entitled "Egr2 and 3 control inflammation, but maintain homeostasis, of PD-1high memory phenotype CD4 T cells". We would be happy to publish your paper in Life Science Alliance pending final revisions necessary to meet our formatting guidelines.

- please send an enhanced rebuttal that clearly shows for each point where in the manuscript changes were made to address the points as highlighted in the rebuttal. It is helpful to highlight the changes in yellow in the manuscript also.
- please update email address for author TieShang Wang; the email sent to the current email address bounced back
- please have all 3 corresponding authors add their ORCID ID. You should have received instructions on how to do so.
- please list 10 authors et al. in your references

A. FINAL FILES:

B. MANUSCRIPT ORGANIZATION AND FORMATTING:

Sincerely,

Reilly Lorenz
Editorial Office Life Science Alliance
Meyerhofstr. 1
69117 Heidelberg, Germany
t +49 6221 8891 414
e contact@life-science-alliance.org
www.life-science-alliance.org

Editor

Life Science Alliance

Dear Editor

We would like to thank the reviewers for their valuable comments and suggestions regarding our manuscript (LSA-2020-00766-T). We have addressed these points and added additional data into the revised manuscript. Our responses to the comments are as follows:

Reviewer #1 (Comments to the Authors (Required)):

Symonds et al detail the effects of Egr2/3 expression on homeostasis, gene expression, and cytokine production by memory-phenotype (MP) PD1+CD4+ T cells. Their work nicely shows that the expression of Egr2/3, while not essential for generation of the PD1+ (and Lag3+) phenotype, does have an important role in both promoting survival/proliferation and suppressing cytokine secretion by these MP T cells. The latter effects may in part be due to effects of Egr2/3 on regulation of Tbet expression. Overall, the work is convincing and seems to be well controlled and presented. However, a few clarifications/additions would improve accessibility or impact of the work.

Specific comments:

Fig1E is cited in 1st section of results but there is not panel E in fig 1. Also, it seems unsurprising that only mice with the Egr2-gfp cells stain positive for Egr2-gfp (Fig. 1D). Unclear what the point of this panel is.

The erroneous reference to Fig 1E (page 5) has now been removed. We previously showed that Egr2 expression in effector T cells is transient and abolished upon transfer to uninfected mice (Miao et al., 2017). In contrast, Fig 1D shows that Egr2 expression in MP T cells is maintained upon adoptive transfer. We have now expanded upon this in the text to emphasize this point (page 5).

Fig 2 states that Ki67 staining was evaluated in mice that were 7 weeks old. How long was this after reconstitution? Also, the transfers were into wt mice. What is driving the homeostatic proliferation here?

Ki67 was analysed between 8 to 12 weeks after reconstitution. This information has been added to the figure legend (page 37). In wild type mice, homeostatic proliferation is largely driven by homeostatic cytokines such as IL-7 (Boyman et al., 2007; Raeber et al., 2018).

Fig 3B - the color scale for P values here is not legible. Suggest labeling rows individually. Is there differential expression of IL2/15Rb? Perhaps this could explain the effects on proliferation/homeostasis.

Fig 3B has been improved to make it clearer. Il2rb is not differentially expressed in Egr2/3-/- or GFP-Egr2- MP cells compared to GFP-Egr2+ MP cells in the RNAseq.

Fig 5 J/K indicate increased IFNg production by EGR2/3-/- OTII cells after infection and data in Fig 7 further support a suppressive effect of Egr2/3 expression of IFNg in response to IL-12. Presumably the effects of Egr 2/3 on IFNg are secondary to effects on tbet (Tbx21). However, only a subset of the Egr2/3- MP T cells are producing IFNg. Are the remaining T cells not responding to IL-12 (IL12R or

STAT4 neg)? Do they instead produce more IL-10 (expected from RNA) or other effector cytokines (IL-17, IL-4, etc) during infection, in response to IL-12, or when stimulated in other contexts?

We and others previously found that only a subset of T cells produce IFN γ during infection (Miao et al 2017, Roman et al 2002, Foulds and Shen 2006) and we have now highlighted this in the text (page 10). The reasons for this are unclear but may reflect different microenvironments. As the reviewer points out, the effects on IFN γ are likely secondary to effects on T-bet as we published previously (Singh et al., 2017). As we have shown in Fig. 7C and D, around 40-45% of Egr2/3^{-/-} MP cells express T-bet while the expression of IFN γ by Egr2/3^{-/-} MP during infection is only slightly lower (30-40%). We did not find increased expression of Il17 or Il4 by Egr2/3^{-/-} MP cells in the RNAseq. Although it will be interesting to investigate the differences between Egr2/3^{-/-} and Egr2⁺ MP cells in cytokine production and other functions under different conditions, in this study we are focussing on homeostatic conditions.

It seems important to also comment on whether the proportion of FoxP3⁺ CD4 T cells reduced in the Egr2/3^{-/-} mice, which might also promote an inflammatory response.

We previously showed that Treg numbers and function were unchanged in Egr2/3^{-/-} mice (Li et al 2012). In addition, we have now added data into the revised manuscript showing that the proportions of Tregs amongst Egr2/3^{-/-} and Egr2⁺ CD44^{high} cells are similar (page 5, revised Fig 1B).

Reviewer #2 (Comments to the Authors (Required)):

In the manuscript "Egr2 and 3 control inflammation, but maintain homeostasis, of PD-1^{high} memory phenotype CD4 T cells", Symonds and Zheng et al investigate the mechanism by which Egr2 and 3 control autoimmune activity by memory phenotype (MP) cells. They demonstrate, using GFP-Egr2 knockin and Egr2/3 conditional knockout mice and an OT-II retrogenic model, that while Egr2 and Egr3 are not required for formation of PD1^{high} MP cells, they are required for proliferation and homeostasis of these cells, promote a distinct gene expression pattern, and are required to control inflammatory responses by these cells. They also analyzed these cells in rheumatoid arthritis patients and found reduced Egr2 associated with inflammatory responses in RA.

The manuscript is clearly written. A major strength of this study is use of several genetically modified mouse lines and mixed bone marrow chimeras to ensure study of cell-intrinsic effects of EGR2 on PD1^{hi} MP cells. Another strength is correlation of these results to data from human patients. These findings take important steps towards elucidating the mechanism by which MP cells contribute to autoimmunity. However, there is some concern regarding the use of Egr2/3 double conditional knockout as a control for the gene expression analysis. Specifically, Egr2 and Egr3 have distinct roles in T cells as evidenced in part by a previous publication in which more severe autoimmunity was found in the absence of both (Morita et al 2016). As such, it is not clear why the authors chose to determine EGR2 regulated genes using the double knockout for the RNAseq portion of the analysis rather than an EGR2 knockout and how to interpret these genes as being directly regulated specifically by EGR2.

We and others have previously demonstrated that Egr2 and Egr3 have overlapping functions in T cells, but Egr2 is dominant in all the functions analysed (Li et al 2012, Morita et al 2016, Miao et al 2017). Autoimmunity is not observed in Egr3 knockout while CD2-Egr2^{-/-} only develops autoimmunity in later life (Tourtellotte et al 1998, Zhu et al 2008, Li et al 2012). However, the severe autoimmunity and loss of homeostasis of T cells is only seen in Egr2/3^{-/-} mice (Zhu et al, 2008; Li et al, 2012; Morita et al 2016) as the reviewer points out. Therefore, we utilised Egr2/3^{-/-} cells for RNAseq to avoid compensation by Egr3. We did not conclude that the genes were regulated

specifically by Egr2 and not Egr3; indeed, previous reports have indicated shared target genes for Egr2 and Egr3 such as FasI (Rengarajan et al 2000). We have now changed the text to emphasize further that we are examining the function of both Egr2 and Egr3 in the RNAseq (pages 6 and 7).

Additional minor issues:

1. The conclusion that Egr2/3 support self-renewal of PD-1^{high} MP cells appears to be based primarily on the fact that Egr2 levels remain constant with cell divisions. It would be more appropriate to either modify the conclusion to focus on homeostatic expansion OR include further evidence that the population after cell divisions is the same cell population. Specifically, data demonstrating the same phenotype across multiple markers for the undivided and most divided peaks would provide further evidence towards this point.

We have now changed the text to refer to homeostatic proliferation rather than self-renewal (pages 5, 6, 15, 17).

2. For differential gene expression in Fig. 3A, the list of genes listed in the text description includes genes not listed on the figure. It would help the clarity for the reader if the list of genes in the text and in the figure were the same.

We have now improved Fig 3A to better reflect the text as the reviewer suggested.

3. The discussion of Egr2/3 targets and genes associated with proliferation defects in results for Fig. 4E,F does not specify which genes are upregulated or downregulated with proliferation defects. Detail on upregulation vs downregulation would add clarity for the reader.

We have now added this to the text to make it clearer (page 8).

4. It is not entirely clear what each of the top three rows for each gene in Figure 4F represent - they are clearly chromatin peaks, but it's unclear what the comparison is between the three. And how are RNA-seq reads incorporated in this figure?

The first two rows are the RNAseq reads for that gene for Egr2⁺ and Egr2/3^{-/-} MP cells, while the third is the peaks from GFP-Egr2 ChIPseq. We have now improved the figure legend to make this clearer (page 39).

5. When Figure S2 is referenced, clarity would be enhanced by citing the subsections for each point of discussion.

Citation of subsections has now been added to the text (page 9).

6. Figure 6 depicts analysis of CD127 through representative data. Providing the data across all samples would provide statistical justification for your conclusion.

We have now performed a statistical analysis of CD127 which shows that the percentage of CD127^{high} cells is similar between Egr2⁺ MP and Egr2/3^{-/-} MP cells. These data have now been included in revised Fig 6B.

7. The analysis in Figure 6 and accompanying text for CD127 levels suggest that the cells respond to IL7 based on receptor expression, but this is not proven. It would improve this section to either temper the conclusion so that response to IL-7 is not the focus, or to complete an experiment stimulating the cells in vitro through CD127 and measuring activation of downstream signaling.

We have now reworded this section accordingly (page 10).

8. Did the healthy controls in Fig. 8C,D also have >10% T-bet+? If not, how were they selected? More broadly, to the reader Fig. 8A, B are very strong points that are well suited to a strong ending. C and D may dilute the point, and therefore might be more appropriate as supplemental data.

The healthy controls in Fig 8C had percentages of T-bet+ cells close to the average for the healthy controls (~5%). The statistical data in Fig 8D are comparing T-bet expression in Egr2+ and Egr2- cells from patients only. We have now improved the figure legend to make this clearer (page 41). We think antagonising T-bet function is one of the major roles of Egr2. Therefore, we will present these in our main text.

Reviewer #3 (Comments to the Authors (Required)):

Symonds et al provide a manuscript describing the roles of EGR2/3 on memory T cell function with a focus on PD-1hi CD4+ T cells. They demonstrate Egr2/3 expression in a Cd4+ memory T cell population enriched for memory and Th1 features. These cells appear proliferative with a distinctive transcriptomic signature. In parallel, Egr2/3-/- cells are found to be abnormal with increased PD1 expression, decreased proliferative capacity, reduced TCR repertoire, and increased Th1 skewing. In RA patients, the expanded PD-1hi T cell population shows reduced Egr2/3 expression, associated with increased Tbet and GZMB expression. The work highlights Egr2/3 as an interesting regulator of CD4 T cell proliferation and function, now placed in the context of pathologically expanded PD1 high T cells in autoimmunity. The work in multiple models, using complementary approaches, with in vivo experiments, transcriptomics, and ChIP analyses, is broad and interesting. It would be valuable for the authors to address these points, in particular with focus on clarifying 1) the relationship between 'PD1hi' cells studied here and the PD1hi cells described in autoimmune patients, and 2) the extent to which PD1+ cells in Egr2/3-/- mice can be compared to the Egr2/3+ PD1+ subset in WT or GFP+ mice.

MAJOR COMMENTS:

1) PD1 expression:

How do the authors distinguish 'PD-1hi' versus PD1+ or PD1intermediate? Figure 1 only indicates a PD1 +/- cutoff. Is this the same gate used to analyze cells in Figure? Does Hi = positive?

Yes, PD-1high is the same as PD-1+ and the same gate was used as in this figure. We have now included this information in the text (page 5).

2) Experimental numbers: Figure 1,2,

The total numbers of mice should be indicated more clearly.

For example, Figure 1C/D. How many mice are analyzed here? The legend indicates 3 experiments, but there are 4 data points. Are the multiple mice per group per experiment?

Three experiments with four to five mice in each group in each experiment. The data in Fig 1D are from four mice from one experiment. This information has been added to the respective figure legends (pages 37 and 38).

3) Egr2/3-/- cells

It is not clear how to think about the T cells in the Egr2/3-/- mice. They all express PD1, yet they look like functionally and transcriptomically seem to resemble PD-1- EGR2/3- cells in WT mice. Do you the authors have evidence to suggest the the PD1+ cells in Egr2/3-/- are phenotypically like PD-1hi cells

described in disease (e.g. the Tfh-like cells in Rao/Zhang/Bocharnikov)? The authors could discuss further what they think of this T cell phenotype in the absence of Egr2/3. They seem unlikely to be the same T cell population as the PD1hi subset in WT mice.

The papers showing enrichment of pathogenic PD-1hi T cells in RA and SLE have found that these cells do not express the Tfh marker CXCR5 but instead express CXCR3 and CCR5 (Rao et al, 2017; Zhang et al, 2019; Bocharnikov et al, 2019; Caielli et al, 2019). These markers are highly expressed by Egr2+ MP cells and also by Egr2/3-/- MP cells (Fig. 1B) and we have now highlighted the similar phenotypes of these cells in the discussion (page 15). In terms of these cell surface markers, Egr2/3-/- MP cells more closely resemble Egr2+ MP cells than Egr2- MP cells (Fig. 1B). We hypothesise that this population is normally controlled by Egr2 and the absence of Egr2 and 3 in the Egr2/3-/- cells leads to them acquiring some functional and transcriptomic features of the Egr2- MP population. This hypothesis will be investigated further in future studies.

4) They authors have not clearly demonstrated the that PD1hi cells they analyze are functionally similar to PD1hi Tfh-like cells in RA patients (studies they reference). It still seems possible that the PD1hi cells studied in this report are Tregs or other activated cells unrelated to Tfh cells. Can they authors demonstrate a Tfh-like function, or specific enrichment in Tfh-like features, in the 'PD1hi' cells they study?

The pathogenic PD-1hi T cells which have been discovered in RA and SLE are distinct from Tfh cells and do not express the Tfh marker CXCR5; instead they express CXCR3 along with high levels of Il21 and Il10 (Rao et al, 2017; Zhang et al, 2019; Bocharnikov et al, 2019; Caielli et al, 2019). Il21 and Il10 have been reported to be important for the extrafollicular B cell helper function of these cells (Bocharnikov et al, 2019; Caielli et al, 2019). These cytokines are highly expressed by Egr2/3-/- MP cells (Fig 3A), which we have now highlighted in the discussion (pages 15 and 16).

5) Egr+ cells

The transcriptomics suggest high expression of FoxP3 and CD25 in the Egr2/3+ cells. Are they enriched Tregs? If Egr is induced by TCR activation, then are the Egr+ cells in the GFP mice largely an activated T cell population?

We have now added data showing that although some CD44high Egr2+ cells are indeed Tregs, the majority are not and, importantly, the Egr2/3-/- CD44high population had a similar proportion of Tregs (page 5, revised Fig 1B).

Egr2 is indeed induced by TCR activation in vitro and during viral infection but the Egr2+ MP cells lack expression of activation markers such as CD25 (revised Fig 1B).

6) In the adoptive transfer experiments with OTII mice, did the authors evaluate if Egr2/3 loss changes the migration of the cells, such that they migrate to different tissues? differential expression of CCR9, ICAM, etc could reflect changes in migratory capacity.

We did not analyse the migration of OT-II cells. However, in our previous experiments with OT-I effector T cells we found that the numbers of cells in infected tissues were also reduced indicating that the observed impairments in expansion of Egr2/3-/- cells are not due to increased migration (Miao et al 2017). In addition, we have now discussed the issue of the high levels of chemokine receptors on Egr2/3-/- MP cells in the discussion (page 16).

MINOR:

1) the "MP" abbreviation is non-standard and not that helpful. The authors could describe the cells

as a memory T cell subset and then omit the MP label throughout the manuscript to improve readability, or simply use the term 'memory'.

The term MP has been used by several groups (e.g. Younes et al., 2011; Su et al., 2013; Kawabe et al., 2017).

2) It appears that all CD44^{hi} cells are CCR5⁺ and the majority are CXCR3⁺. This seems unexpected; how do the authors explain this staining pattern?

CCR5 and CXCR3 are known to be preferentially expressed by MP cells (Bleul et al 1997, Oberbarnscheidt et al 2011, Li et al 2016) and it has been previously reported that the majority of MP cells are CXCR3⁺ (Sallusto et al 1998), which we have now pointed out in the text (page 16). We do find high expression of CCR5 on CD44^{high} cells in our models; however, the expression is similar between GFP-Egr2 and Egr2/3^{-/-} MP cells.

3) It is not clear what the authors mean with the term 'virtual' MP T cells or 'virtual' PD-1^{hi} cells

The term 'virtual' was used for the OT-II cells since these cells are antigen inexperienced; we have now changed the text to 'antigen-inexperienced' instead to make this point clearer (pages 9, 39).

July 10, 2020

RE: Life Science Alliance Manuscript #LSA-2020-00766-TRR

Prof. Ping Wang
Barts and The London School of Medicine and Dentistry, Queen Mary University of London
The Blizard Institute
4 Newark Street
London E1 2AT
United Kingdom

Dear Dr. Wang,

Thank you for submitting your Research Article entitled "Egr2 and 3 control inflammation, but maintain homeostasis, of PD-1high memory phenotype CD4 T cells". It is a pleasure to let you know that your manuscript is now accepted for publication in Life Science Alliance. Congratulations on this interesting work.

DISTRIBUTION OF MATERIALS:

Again, congratulations on a very nice paper. I hope you found the review process to be constructive and are pleased with how the manuscript was handled editorially. We look forward to future exciting submissions from your lab.

Sincerely,

Reilly Lorenz
Editorial Office Life Science Alliance
Meyerhofstr. 1
69117 Heidelberg, Germany
t +49 6221 8891 414
e contact@life-science-alliance.org
www.life-science-alliance.org